# DNGR-1 signalling limits dendritic cell activation for optimal antigen cross-presentation

Michael D Buck [1✉], Tomás Castro-Dopico [1], Oliver Schulz[1], Ana Cardoso [1,6], Probir Chakravarty[2], Nathalie Legrave [3,7], Conor M Henry[1,8], Johnathan Canton[1,9], Estelle Wu [1], Sonia Lee[1], Neil C Rogers [1], Enzo Z Poirier [1,10], William Stainier [1], Victor Bosteels[1], Eleanor Childs[1], James I MacRae [3], J Mark Skehel[4], Santiago Zelenay [5] & Caetano Reis e Sousa [1✉]

## Abstract

**Innate immune receptors often induce activation of conventional dendritic cells (cDCs) and enhance antigen (cross-)presentation, favouring immune responses. DNGR-1 (CLEC9A), a receptor expressed by type 1 cDCs (cDC1s) and implicated in immune responses to viruses and cancer, recognises F-actin exposed on dead cell remnants and promotes cross-presentation of associated antigens. Here, we show that recruitment of phosphatase SHIP1, a process governed by a single amino acid residue adjacent to the signalling motif of the receptor, partly explains how DNGR-1 fails to trigger cDC1 activation in vitro. Substituting this residue converts DNGR-1 into an activating receptor but decreases induction of cross-presentation of dead cell-associated antigens. Introducing the reverse mutation into the related receptor Dectin-1 impairs its activation capacity while enhancing its ability to promote cross-presentation. These findings reveal a functional trade-off in receptor signalling and suggest that DNGR-1 has evolved to prioritise antigen cross-presentation over cellular activation, possibly to minimise inflammatory responses to dead cells.**

**Keywords** DNGR-1; CLEC9A; cDC1; Activation; Cross-presentation
**Subject Categories** Immunology; Signal Transduction

## Introduction

Conventional dendritic cells (cDCs) act as sentinels of the immune system by capturing, processing, and presenting antigens to T lymphocytes, as well as transmitting information to other leukocytes (Cabeza-Cabrerizo et al, 2021; Norbury et al, 2002). Among the various cDC subsets, type 1 cDCs (cDC1s) play a

central role in mounting immunity against viruses and tumours (Bottcher and Reis e Sousa, 2018; Huang et al, 1994; Pittet et al, 2023; Sigal et al, 1999; Smed-Sorensen et al, 2012; Wculek et al, 2020). This specialised function is partly attributable to their ability to cross-present exogenous antigens derived from dead cell debris on MHC class I (MHC-I) molecules to activate cytotoxic CD8[+] T cells (Heath et al, 2004; Hildner et al, 2008; Iyoda et al, 2002; Schulz and Reis e Sousa, 2002; Theisen et al, 2018; Yewdell et al, 1988). In addition, cDC1s are key mediators of immune homeostasis and shape both innate and adaptive immune responses through the detection of pathogen and damage-associated molecular patterns (PAMPs and DAMPs, respectively) (Gong et al, 2020; Ma et al, 2024; Takeuchi and Akira, 2010). This is governed by their expression of an array of innate immune receptors, including members of the Toll-like (TLR), NOD-like (NLR), RIG-I-like (RLR), Fc (FcR), and C-type lectin receptor (CLR) families (Bruhns and Jonsson, 2015; Burberry et al, 2014; Geijtenbeek and Gringhuis, 2009; Loo and Gale, 2011; Nimmerjahn and Ravetch, 2008; Reis e Sousa et al, 2024). Triggering of these receptors can lead to cellular activation, whereby cDC1s exit a state of quiescence and surveillance, display increased migratory activity towards T cell areas of lymphoid tissues and become competent at priming naïve T cells and directing their effector differentiation (Cabeza-Cabrerizo et al, 2021). Marked changes in gene expression and cellular metabolism often accompany this activated state and underlie the production of costimulatory molecules, migratory receptors, and secreted products required for T cell stimulatory functions (Buck et al, 2017; O'Neill and Pearce, 2016; Pearce and Everts, 2015).

DNGR-1 (also known as CLEC9A) is a cDC1-specific type II transmembrane CLR that mediates cDC1 recognition of dead cell debris through binding to F-actin exposed on cell corpses (Ahrens et al, 2012; Hanc et al, 2015; Huysamen et al, 2008; Iborra et al, 2012; Sancho et al, 2009; Zelenay et al, 2012; Zhang et al, 2012). DNGR-1 is an endocytic receptor that can mediate uptake of F-actin-coated beads but is not absolutely required for the

[1]Immunobiology Laboratory, The Francis Crick Institute, London, UK. [2]Bioinformatics and Biostatistics, The Francis Crick Institute, London, UK. [3]Metabolomics, The Francis Crick Institute, London, UK. [4]Proteomics, The Francis Crick Institute, London, UK. [5]Cancer Inflammation and Immunity Group, Cancer Research UK Manchester Institute, The University of Manchester, Manchester, UK. [6]Present address: Medical Department, ADM Health & Wellness, London, UK. [7]Present address: Metabolomics Platform, Luxembourg Institute of Health, Strassen, Luxembourg. [8]Present address: Roche Products Ltd., Welwyn Garden City, UK. [9]Present address: Comparative Biology and Experimental Medicine, Faculty of Veterinary Medicine, University of Calgary, Calgary, AB, Canada. [10]Present address: Innate Immunity in Physiology and Cancer Laboratory, Institut Curie, PSL Research University, INSERM, Paris, France. ✉E-mail: michael.buck@crick.ac.uk; caetano@crick.ac.uk

internalisation of dead cell debris (i.e. necrophagy), possibly because of redundancy with other necrophagy receptors (Canton et al, 2021; Schulz et al, 2018). Instead, DNGR-1 plays a non-redundant role post-necrophagy, favouring cross-presentation of antigens associated with internalised dead cell debris. Indeed, DNGR-1 triggering in phagosomes by F-actin exposed by dead cell debris can lead to phagosomal membrane destabilization and rupture (Canton et al, 2021), thus allowing dead-cell derived antigens to access the cDC1 cytosol and enter the endogenous MHC-I processing and presentation pathway (Colbert et al, 2020; Cruz et al, 2017; Gros and Amigorena, 2019). Importantly, expression of DNGR-1 in heterologous cells (both immune and non-immune) can enhance cross-presentation of ligand-associated antigens (Canton et al, 2021). This demonstrates that cross-presentation of dead cell-associated antigens is not always an intrinsic cell biological property of cDC1s but one that can be induced by signalling from cDC1-restricted receptors. Thus, understanding the regulation of cross-presentation by cDC1s in part requires the studying of such receptor signals.

DNGR-1 signals via SYK, which is recruited to the cytoplasmic domain of the receptor upon phosphorylation by SRC family kinases of a key tyrosine within the hemITAM motif (Henry et al, 2023a; Mocsai et al, 2010). Other CLRs, such as Dectin-1, a receptor for yeast and bacterial β-glucans, also possess a tyrosine-containing hemITAM motif and similarly signal via SYK to induce robust activation of myeloid cells, including cDCs (Leibundgut-Landmann et al, 2008; Rogers et al, 2005; Underhill et al, 2005). This occurs via stimulation of multiple signalling pathways downstream of SYK, including phospholipase C gamma (PLCγ), MAPK and NF-κB cascades, which coordinately lead to induction of many genes encoding proinflammatory mediators, as well as a shift towards glycolytic cellular metabolism that supports proinflammatory function (Bauer and Steinle, 2017; Dominguez-Andres et al, 2017; Thwe et al, 2019). Interestingly, while exposure of cDC1s to dead cell debris in vitro or in vivo can induce changes in gene and protein expression (Bosteels et al, 2023), these are not affected by DNGR-1 deficiency (Zelenay et al, 2012). Although this could reflect redundancy with other dead cell receptors, as seen in necrophagy analyses, it could also be that DNGR-1 signalling, unlike that by Dectin-1, might be intrinsically unable to activate cDC1s (Zelenay et al, 2012). Conversely, Dectin-1 signals induce only limited cross-presentation of β-glucan-associated antigens (Canton et al, 2021). Here, we uncover a fundamental dichotomy in the ability of DNGR-1 and Dectin-1 to signal for phagosomal damage and cross-presentation versus cellular activation. We show that these two effector functions in DNGR-1 are governed by a single amino acid adjacent to the hemITAM. Conservation of this amino acid across species suggest that the inability of DNGR-1 to signal for activation may have been evolutionarily selected to limit auto-inflammatory or autoimmune responses to cell death.

# Results

## DNGR-1 signalling induces cross-presentation of ligand-associated antigens but does not activate cDC1s

Using primary cDC1s isolated from FLT3L-supplemented bone marrow cultures (Fig. EV1A,B), we first confirmed that DNGR-1 deficiency markedly impairs cross-presentation of dead cell-associated ovalbumin (OVA) but does not alter presentation of exogenous pre-processed OVA peptide (SIINFEKL), nor impacts bulk internalisation of dead cell debris (Fig. 1A,B). To systematically investigate the signalling functions of DNGR-1 in cDC1, we utilised an orthogonal approach more amenable to genetic manipulation. The splenic cDC1 cell line MuTuDC1940, henceforth referred to as MuTuDC, expresses DNGR-1 and has been used extensively to investigate cDC1 biology (Canton et al, 2021; Fuertes Marraco et al, 2012). We generated DNGR-1 knockout (KO) MuTuDCs and complemented them with either wild-type (WT) DNGR-1 (C9) or with a DNGR-1 receptor bearing two mutated tryptophan residues (W155A-W250A; KO/C9(2WA)) that is unable to bind ligand (Hanc et al, 2015), or with a signalling-incompetent receptor generated by mutating the key tyrosine in the hemITAM (Y7F; KO/C9(Y7F)) (Sancho et al, 2009) (Fig. 1C). As expected, all cell lines were equally able to present exogenous OVA peptide to OT-I cells, but, compared to C9-expressing MuTuDCs, C9 KO, KO/C9(2WA), or KO/C9(Y7F) MuTuDCs displayed a defect in cross-presentation of dead cell-associated OVA, or beads bearing both DNGR-1 ligand (DNGR-1L) and OVA (Fig. 1D,E). Necrophagy by MuTuDCs was, again, unaffected by DNGR-1 deficiency (Fig. 1F). These data confirm that both DNGR-1 recognition and signalling are required for efficient cross-presentation of dead cell-associated antigens but not for overall dead cell uptake (Canton et al, 2021; Sancho et al, 2009). Recent studies have shown that phagosomal rupture underpins the ability of DNGR-1 to induce cross-presentation (Canton et al, 2021; Henry et al, 2023a). We measured the phagosomal accumulation of lysenin-mCherry, a cytosolic probe that detects phagosomal damage. This probe consists of mCherry fused to a mutant (W20A) lysenin, which has lost its pore forming ability but is recruited from the cytosol to damaged endocytic vesicles through binding to luminal sphingomyelin (Ellison et al, 2020). Using lysenin-mCherry, we confirmed that α-DNGR-1 IgG-coated beads were able to elicit phagosomal damage in MuTuDCs in a DNGR-1-dependent manner (Fig. 1G,H) (Canton et al, 2021; Henry et al, 2023a). Thus, our engineered MuTuDCs recapitulate the biology of DNGR-1-dependent cross-presentation.

To investigate whether DNGR-1 engagement can activate cDC1, we first validated the ability of DNGR-1L to induce robust signalling by measuring activation of a B3Z NFAT reporter cell line expressing DNGR-1 and SYK (Karttunen et al, 1992; Sancho et al, 2009; Schulz et al, 2018). DNGR-1L stimulated reporter activity in a dose-dependent manner with an $EC_{50}$ ~5 nM (Fig. 1I). Similar results were observed when cross-linking the receptor with plate-bound α-DNGR-1 IgG (Fig. 1I). Furthermore, when coupled to beads, α-DNGR-1 IgG effectively led to accumulation of phosphorylated SYK around phagosomes in KO/C9 MuTuDCs (Fig. 1J). Given these results, we compared stimulation of WT or DNGR-1 deficient primary cDC1s and MuTuDCs with DNGR-1L versus canonical innate immune stimuli (Alexopoulou et al, 2001; Montoya et al, 2002). While IFN-α and poly(I:C) robustly induced upregulation of costimulatory markers (CD40, CD86), chemokine receptor CCR7 and (for poly(I:C)) promoted IL-12 p40 production, neither DNGR-1L (Figs. 1K,L and EV1C,D) nor plate-bound α-DNGR-1 IgG (Fig. EV1E) induced any significant changes in activation of primary cDC1s. Similar results were obtained with

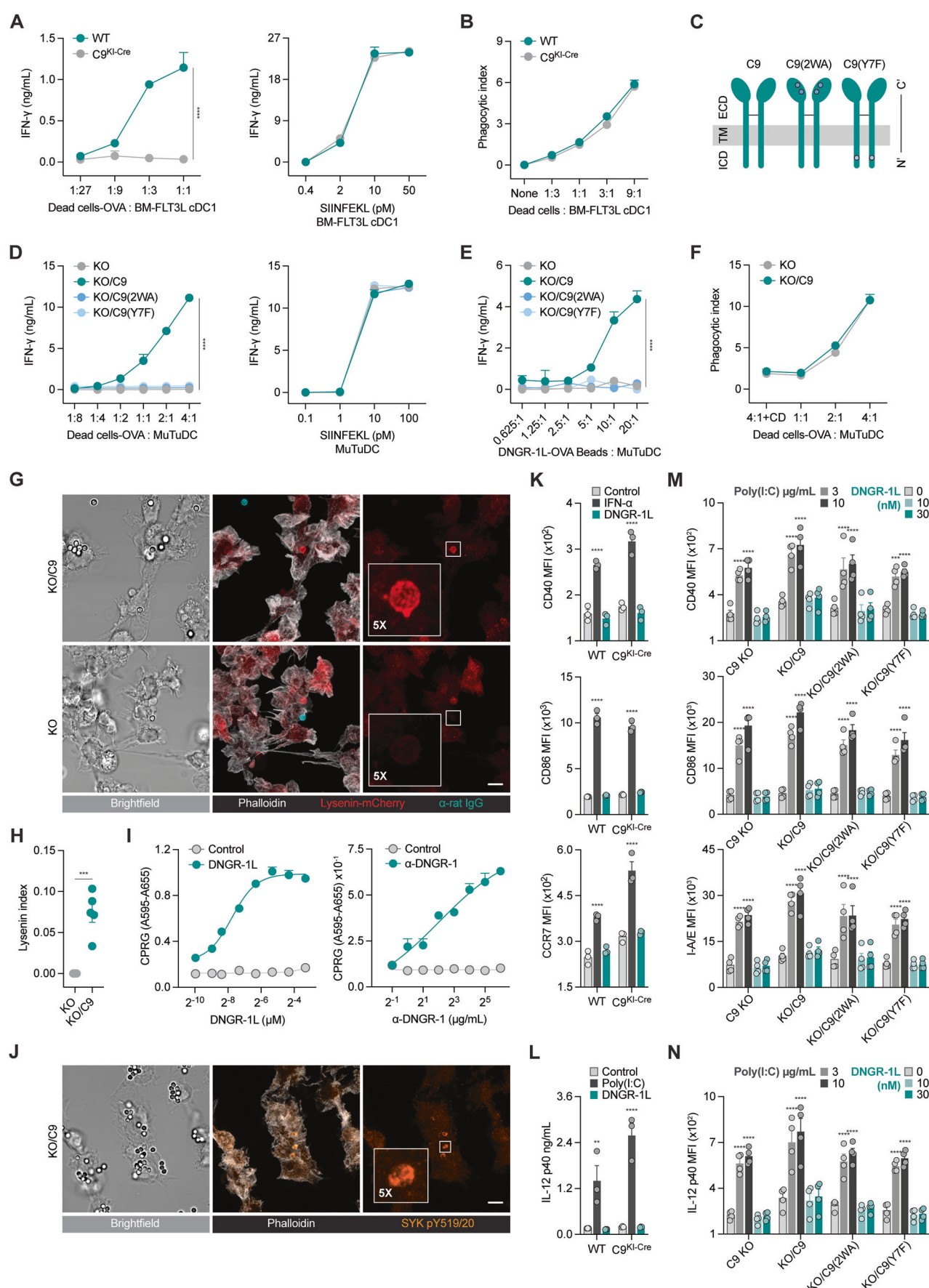

**Figure 1.  DNGR-1 signalling results in phagosomal rupture and cross-presentation of dead cell-associated antigens but does not activate cDC1s.**

(A) ELISA for IFN-γ release from OT-I effector T ($T_E$) cells co-cultured with wild-type (WT) or DNGR-1 deficient *Clec9a* knock-in Cre (C9$^{KI-Cre}$) bone marrow-FLT3L cultured (BM-FLT3L) cDC1s incubated with ovalbumin (OVA)-dead cells (left) or SIINFEKL peptide (right). Mean ± SD from biological duplicates is plotted. (B) Uptake of Cell Tracker-Deep Red (CT-DR)-labelled dead cell debris by WT or C9$^{KI-Cre}$ BM-FLT3L cDC1s assessed by flow cytometry. Plotted as phagocytic index (% CT-DR$^+$ cells x CT-DR MFI of CT-DR$^+$ cells/arbitrary unit) mean ± SD from $n = 3$. (C) Schematic of WT (C9), W155A-W250A (C9(2WA)), or Y7F (KO/C9(Y7F)) DNGR-1 transduced into DNGR-1 knockout (KO) splenic cDC1 line MuTuDC1940 (MuTuDCs). Intracellular cytoplasmic domain (ICD); transmembrane domain (TM); extracellular domain (ECD). (D) ELISA for IFN-γ release from OT-I $T_E$ cells co-cultured with C9 KO, KO/C9, KO/C9(2WA), or KO/C9(Y7F) MuTuDCs incubated with OVA-dead cells (left), SIINFEKL peptide (right), or (E) DNGR-1 ligand (DNGR-1L)-OVA coupled beads. Mean ± SD from biological (D) quadruplet or (E) duplicates is plotted. All lines are plotted even when they cannot be seen because of superimposition. (F) Uptake of CT-DR-labelled dead cell debris as in (B) by C9 KO or KO/C9 MuTuDCs. Cytochalasin D (CD) co-culture was included as a negative control. (G, H) C9 KO or KO/C9 MuTuDCs transduced with lysenin-mCherry fusion protein were co-cultured with α-DNGR-1 IgG coupled beads and assessed by confocal microscopy. (G) Representative images, scale bar = 5 µm. Beads not internalised marked by α-rat IgG staining. (H) Quantification of lysenin-mCherry$^+$ phagosomes per cells in field of view (Lysenin index), bars indicate mean ± SEM. (I) Absorbance of β-galactosidase activity from B3Z-DNGR-1-SYK reporter cells stimulated ± DNGR-1L (left) or plate-bound α-DNGR-1 IgG (right). Mean ± SEM of four replicates. (J) Confocal microscopy of KO/C9 MuTuDCs treated as in (G). (K, L) WT, C9$^{KI-Cre}$ BM-FLT3L cDC1s or (M, N) C9 KO MuTuDCs or cells reconstituted with indicated receptors were cultured overnight ± indicated stimuli and assessed in triplicate by flow cytometry for (K, M) surface or (N) intracellular protein expression, or (L) release of IL-12 p40 by ELISA in cultured supernatants. Cells treated with 200 U/mL IFN-α or 20 nM DNGR-1L (K) or 10 µg/mL Poly(I:C) or 20 nM DNGR-1L (L). Mean ± SEM of each group from biological triplicates (K, L) or pooled duplicates from two independent experiments (M, N) is plotted. MFI mean fluorescence intensity. Data are representative of two (A, B, F–H, J–N) or ≥ three (D, E, I) independent experiments. Data were analysed using Tukey-corrected two-way ANOVA (A–F, K–N) or unpaired *t* test (H). Significant values comparing against C9$^{KI-Cre}$ BM-FLT3L cDC1 (A), C9 KO MuTuDCs (D, E, H), or untreated samples (K–N) are plotted. (A, D, E) ****$P < 0.0001$, (H) ***$P = 0.0002$, (K) ****$P < 0.0001$, (L) **$P = 0.0020$, ****$P < 0.0001$, (M) ***$P = 0.0004$, ****$P < 0.0001$, (N) ****$P < 0.0001$. See also Fig. EV1. Source data are available online for this figure.

MuTuDCs: exposure of KO/C9 cells to DNGR-1L was indistinguishable from the negative controls (no ligand control or DNGR-1L-exposed cells expressing 2WA or Y7F DNGR-1 mutants; Figs. 1M,N and EV1F,G). We conclude that DNGR-1 signalling is essential for the cross-presentation of dead cell-associated antigens but does not activate cDC1s.

## A single amino acid substitution rescues the ability of DNGR-1 to activate cDC1

Dectin-1 and CLEC-2, which are closely related to DNGR-1, have been shown to activate myeloid cells (Brown et al, 2003; Fuller et al, 2007; Goodridge et al, 2007; Gross et al, 2006; LeibundGut-Landmann et al, 2007; Reis e Sousa et al, 2024; Rogers et al, 2005; Zelenay et al, 2012). We compared the cytoplasmic domains of DNGR-1 and Dectin-1 across species (Fig. 2A). Although both receptors possess a hemITAM sequence (Fig. 2B) (Rogers et al, 2005; Severin et al, 2011), the amino acids surrounding the key tyrosine are different. Immediately upstream of the tyrosine in Dectin-1 lies a conserved DEDG sequence that previous studies suggest contributes to activatory signalling (Fuller et al, 2007; Zelenay et al, 2012). In contrast, DNGR-1 possesses an EEEI sequence that is conserved in most species except for mice (AEEI) (Fig. 2B). As the triacidic motif is found in both receptors across most species, we focused instead on the glycine versus isoleucine adjacent to the tyrosine. We hypothesised that cDC1 activation via DNGR-1 might be rescued by replacing the isoleucine with glycine (Fig. 2C). As before, C9 KO MuTuDCs or those expressing C9, C9(2WA) or C9(Y7F) displayed only a marginal increase in CD86 or MHC-II expression in response to DNGR-1L (Figs. 1M,N, 2D,E and EV2A). However, cells expressing C9(I6G) or a chimeric receptor comprising a Dectin-1 (C7) tail with DNGR-1 transmembrane and extracellular domains (KO/C7::C9) robustly upregulated expression of costimulatory molecules and CCR7, as well as secreted chemokines in response to ligand engagement (Figs. 2D,E and EV2A). Expression levels on the cell surface were equivalent for all the receptors tested, arguing for a qualitative rather than quantitative effect (Fig. EV2B).

To determine whether the nature of the extracellular domain had any impact on our observations, in parallel we also generated chimeric receptors bearing the intracellular domains of WT or mutant DNGR-1 with the extracellular domain of Dectin-1, as well as full-length Dectin-1 (Fig. 2C). Constructs were overexpressed in RAW 264.7 cells, which express low levels of endogenous Dectin-1 and have been previously used to study signalling by ectopically expressed receptor (Gantner et al, 2005) (Fig. EV2C). Stimulation with Dectin-1 agonist (hot alkali treated zymosan (Zym-D)) (Underhill et al, 2005) induced signs of activation only in RAW 264.7 cells expressing Dectin-1 or a C9(I6G) tail (Figs. 2F and EV2D), although a modest increase in CCR7 was observed in both KO/C9 MuTuDCs and C9::C7 RAW 264.7 cells (Figs. 2E,F and EV2A,D). Thus, even in a chimeric receptor setting and in a heterologous cell type, substitution of glycine for isoleucine in the cytoplasmic domain of DNGR-1 rescues its ability to mediate myeloid cell activation.

cDC1 activation is accompanied by changes in gene expression that support cytokine production, upregulation of costimulatory molecules, and migration to secondary lymphoid organs (Cabeza-Cabrerizo et al, 2021). We performed bulk RNA sequencing (RNAseq) of KO/C9 or KO/C9(I6G) MuTuDCs treated with DNGR-1L. We found that DNGR-1L profoundly altered gene expression in C9(I6G)-expressing cells but not in cells bearing the WT receptor (C9) (Fig. 2G). Changes included upregulation of costimulatory and migratory molecules (*CD40, CD80, CD83, Icam1*) and cytokines (*Ccl17, Ccl22, Tnf*), which we confirmed by RT-qPCR analysis (Fig. 2H). Gene induction by DNGR-1L in KO/C9(I6G) MuTuDCs was comparable to that in KO/C7::C9 MuTuDCs but was not observed in C9 KO-, KO/C9-, or KO/C9(2WA)-expressing cells (Fig. 2H). This suggested that the I6G DNGR-1 mutant signals in a "Dectin-1-like" manner, a conclusion further supported by analyses showing that gene signatures of Dectin-1 signalling and its associated pathways were significantly enriched in C9(I6G) MuTuDCs after DNGR-1L stimulation (Fig. 2I). Overall, we conclude that the isoleucine adjacent to the hemITAM critically limits the ability of DNGR-1 to signal for cDC1 activation.

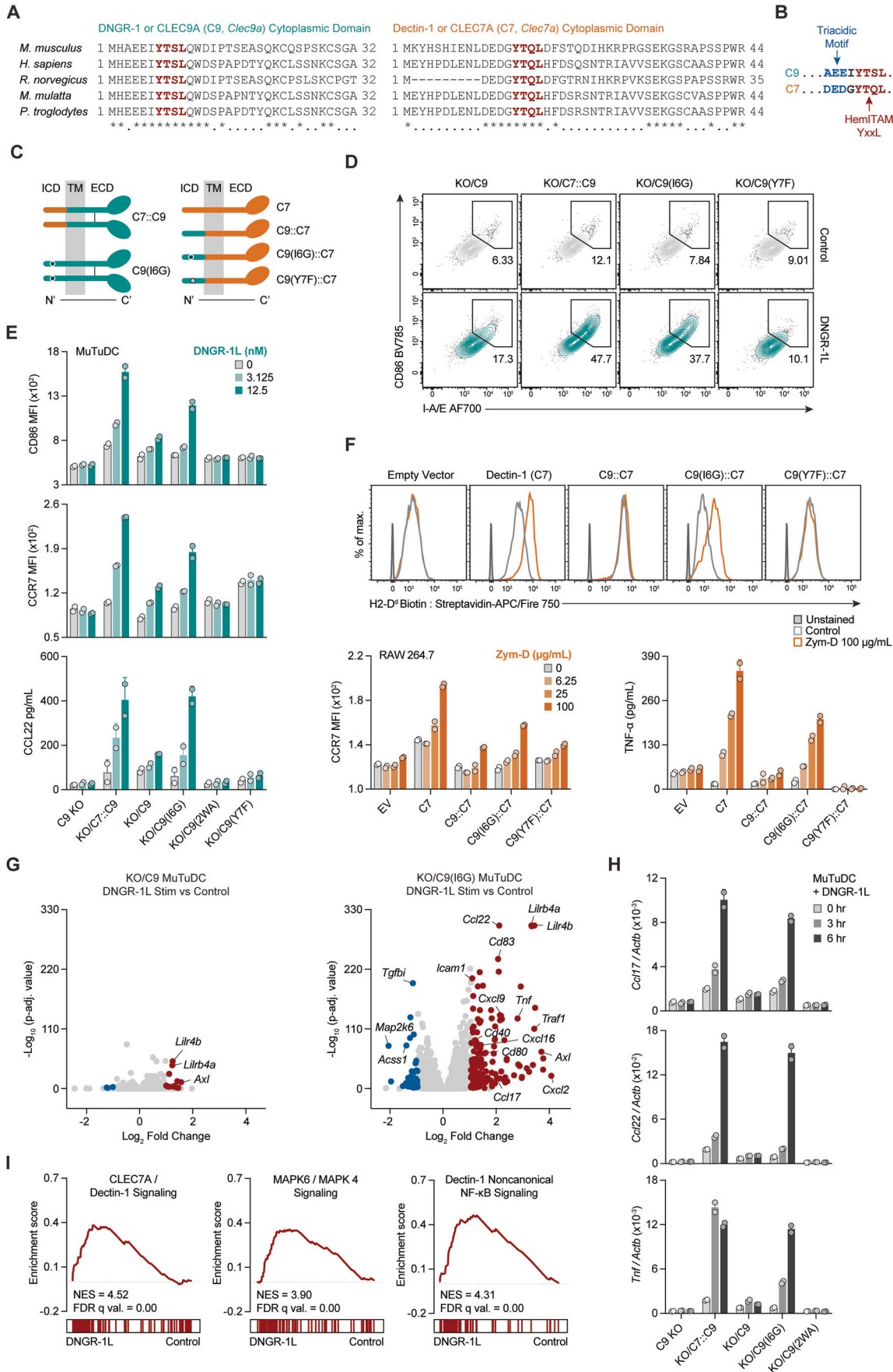

**Figure 2. An isoleucine residue adjacent to the hemITAM motif restrains the ability of DNGR-1 to activate cDC1s.**

(A) Aligned DNGR-1 (CLEC9A, C9; left) or Dectin-1 (CLEC7A, C7; right) cytoplasmic domain sequences from indicated species. HemITAM sequence is bolded in red. Consensus is represented by an asterisk (*) and similarity with a dot (.). (B) Aligned hemITAM motifs of DNGR-1 and Dectin-1 from *Mus musculus*. The triacidic motif from Dectin-1 is also depicted with the corresponding sequence from DNGR-1. (C) Schematic of chimeric receptor comprised of the intracellular cytoplasmic domain (ICD) of Dectin-1 fused with the transmembrane (TM) and extracellular domain (ECD) of DNGR-1 (C7::C9), or I6G DNGR-1 (C9(I6G)) transduced into C9 KO MuTuDC (left). Right side depicts WT Dectin-1 (C7), or chimeric receptor constructs using the TM and ECD of Dectin-1 fused with the ICD of WT (C9::C7), I6G (C9(I6G)::C7), or Y7F (C9(Y7F)::C7) DNGR-1 transduced into RAW 264.7 cells. (D, E) Representative flow cytometric analysis of surface proteins from C9 KO MuTuDCs or those reconstituted with indicated receptors and stimulated overnight ± with DNGR-1L. (E) Quantification of flow cytometric analysis of surface proteins from MuTuDCs in (D), as well as CCL22 in cultured supernatants from those cells. Mean ± SD from biological duplicates is plotted. (F) Representative histograms and quantification of flow cytometric analyses of surface marker MFI or TNF-α in cultured supernatants from RAW 264.7 cells ectopically expressing indicated receptors or transduced with empty vector (EV) stimulated overnight ± with zymosan depleted (Zym-D). Mean ± SD from biological duplicates is plotted. (G) RNAseq volcano plot of differentially expressed genes between 12.5 nM DNGR-1L stimulated versus untreated (Control) KO/C9 (left) or KO/C9(I6G) MuTuDCs (right). N = 3 per group. (H) RT-qPCR analysis of indicated genes from MuTuDCs treated with 12.5 nM DNGR-1L over time. Mean ± SD from biological duplicates is plotted. (I) Gene set enrichment analysis (GSEA) of Reactome signalling pathways identified in DNGR-1L-stimulated KO/C9(I6G) MuTuDCs versus KO/C9 MuTuDCs from (G). Data are representative of two (F, H), or three (D, E) independent experiments. See also Fig. EV2. Source data are available online for this figure.

## DNGR-1 signalling does not induce a glycolytic switch in cDC1s

Activation of myeloid cells via innate immune receptors often triggers rapid changes in cellular metabolism characterised by increased rates of aerobic glycolysis. Glycolytic reprogramming supports both the energetic and anabolic nutrient demands associated with cDC1 activation and migration (Amiel et al, 2012; Buck et al, 2017; Everts et al, 2014; Everts et al, 2012; Guak et al, 2018; Krawczyk et al, 2010; O'Neill and Pearce, 2016; Pearce and Everts, 2015). Additionally, previous studies report that human and mouse mononuclear phagocytes stimulated with β-glucan increase glycolysis in a Dectin-1-SYK-dependent manner (Dominguez-Andres et al, 2017; Thwe et al, 2019). In support of this, glucose metabolism and glycolysis were among the metabolic pathways significantly enriched in the RNAseq dataset from DNGR-1L stimulated KO/C9(I6G) MuTuDCs (Fig. 3A,B). Extracellular flux analysis (EFA) of MuTuDCs (Fig. 3C,D) and RAW 264.7 cells (Fig. 3E) expressing cytoplasmic domain variants of DNGR-1 or DNGR-1::Dectin-1 chimeric receptors confirmed this notion. Extracellular acidification rates (ECAR), a surrogate readout for glycolysis, rapidly increased above baseline only in cells expressing C7 or C9(I6G) cytoplasmic tails, while those expressing C9 or C9(Y7F) tails exhibited no change in response to receptor stimuli. The oxygen consumption rate (OCR), an indicator of oxidative phosphorylation (OXPHOS), did not differ among genotypes (Fig. EV3A,B). Our results therefore indicate that WT DNGR-1 does not signal for an increase in glycolysis upon ligand engagement, which might contribute to its inability to trigger cDC1 activation.

To determine the breadth of metabolic pathways potentially modulated by DNGR-1 signalling (Fig. 3F), we assessed metabolite abundance in KO/C9, KO/C9(I6G), and KO/C7::C9 MuTuDCs after DNGR-1L stimulation by liquid and gas chromatography-mass spectrometry (LC- and GC-MS). Consistent with the EFA data, both C9(I6G)- and C7::C9-expressing MuTuDCs were enriched for all glycolytic metabolites within an hour of DNGR-1L stimulation (Fig. 3G). In contrast, WT cells had enhanced abundance of glycolytic intermediates DHAP and PEP and offshoot glycerol 3-phosphate, but not downstream metabolites pyruvate or lactate. To follow up on these findings, we carried out $^{13}$C-glucose tracing into downstream metabolites by LC- and GC-MS. Indicative of augmented glycolysis and in line with the EFA results

(Fig. 3C–E), the fractional contribution of $^{13}$C into pyruvate and lactate increased twofold to threefold following DNGR-1L stimulation in C7::C9- and C9(I6G)-expressing MuTuDCs (Figs. 3H and EV3C). In contrast, DNGR-1L treated KO/C9 MuTuDCs minimally catabolised labelled glucose into pyruvate or lactate over unstimulated cells. However, they incorporated significantly more $^{13}$C into PEP compared to KO/C7::C9- or KO/C9(I6G)-expressing cells (Figs. 3H and EV3C). Altogether, these data suggest that the activity of pyruvate kinase M2 (PKM2), a rate-limiting glycolysis enzyme that mediates conversion of PEP to pyruvate (Fig. 3F), might be blunted after DNGR-1 signalling.

LPS-induced activation of GM-CSF-cultured bone marrow cells, which have historically been used to model DCs, is PKM2 dependent (Jin et al, 2020; Liu et al, 2018). We reasoned that blunting PKM2 activity might therefore inhibit cDC1 activation by C9(I6G) or C7::C9 signalling, whereas augmenting PKM2 activity might rescue activation by WT DNGR-1. To test this pharmacologically, we first confirmed that treatment of MuTuDCs with C3K, a PKM2 inhibitor (Ning et al, 2017), decreased their glycolytic capacity (maximal ECAR after mitochondrial function inhibition) in a dose-dependent manner without affecting basal ECAR and OCR (Fig. 3I). On the other hand, acute or prolonged treatment with DASA-58, a PKM2 agonist (Anastasiou et al, 2012), increased ECAR over time while leaving OCR and the glycolytic capacity mostly unperturbed (Figs. 3J and EV3D). Concomitant treatment of MuTuDCs with DNGR-1L and C3K, blunted cDC1 activation in C7::C9- and C9(I6G)-expressing cells, confirming that PKM2 is required for cDC1 activation (Fig. 3K). However, increasing PKM2 activity with DASA-58 failed to restore cDC1 activation downstream of DNGR-1 (Fig. 3K). Together, these data suggest that while enhanced PKM2 activity is required for cDC1 activation, it is not sufficient to rescue activation downstream of DNGR-1 signalling.

## cDC1 activation by DNGR-1 is curtailed by SHIP1

To uncover other factors that might limit the ability of DNGR-1 to signal for cDC1 activation, we undertook an unbiased proteomics approach. Lysates from C9 KO MuTuDCs were incubated with biotinylated peptides corresponding to the cytoplasmic domain of WT and I6G DNGR-1, either as non-phosphorylated or tyrosine-phosphorylated (7pY) versions. Proteins associating with biotinylated DNGR-1 peptides were pulled-down with streptavidin beads

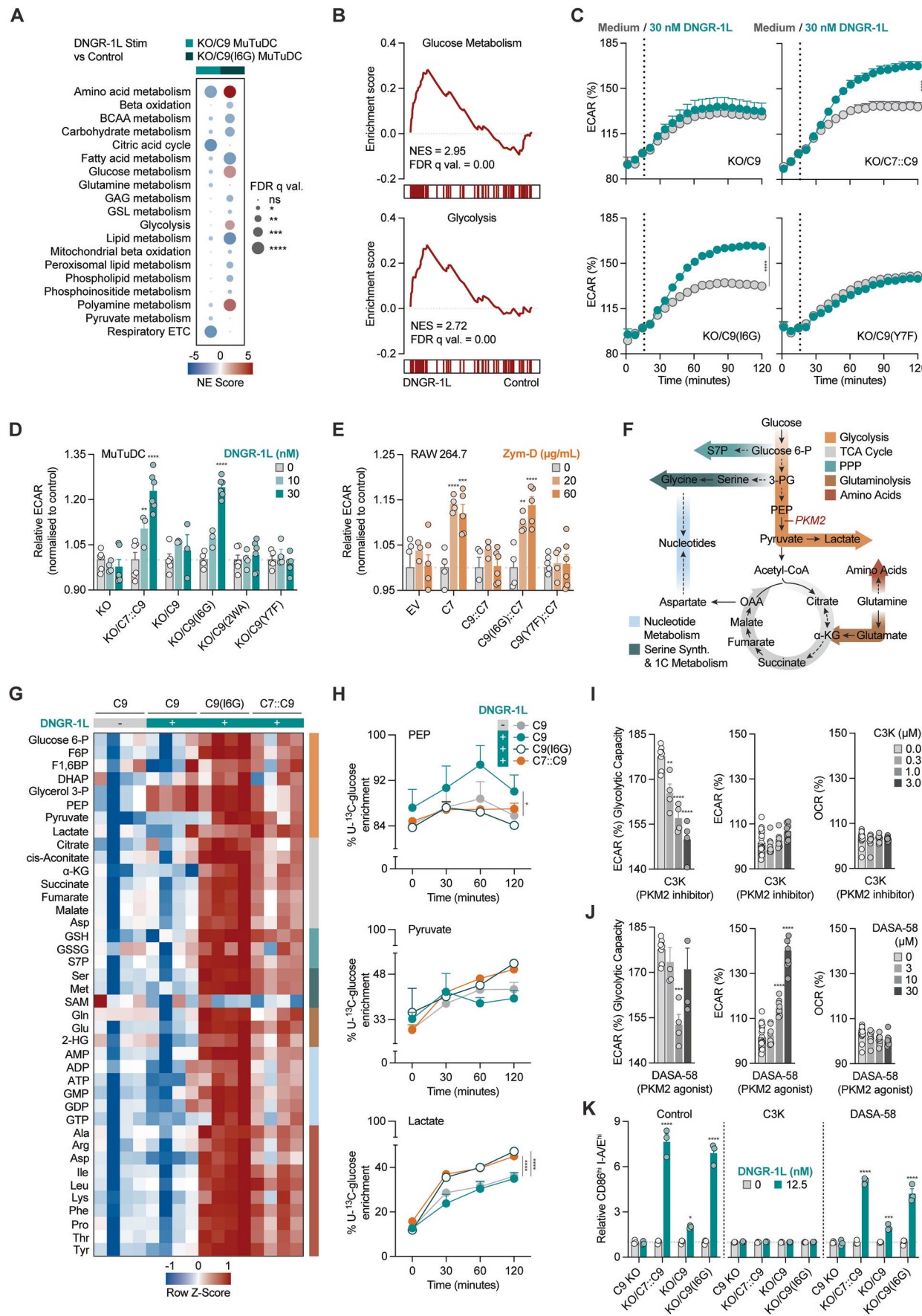

**Figure 3. cDC1 glycolytic switch is restricted during DNGR-1 signalling.**

(A) GSEA of Reactome metabolic pathways identified in C9 KO MuTuDCs reconstituted with C9 or C9(I6G) ± DNGR-1L stimulation. (B) GSEA of Reactome glucose metabolism and glycolysis pathways in KO/C9(I6G) MuTuDCs from (A). (A, B) Data derived from experiment in Fig. 2G. (C) Extracellular acidification rate (ECAR, indicator of glycolysis) measured at baseline and after ± 30 nM DNGR-1L injection of C9 KO MuTuDCs reconstituted with indicated receptors. $N = 3$–6 per group. Data normalised to baseline measurement immediately after injection with stimuli and shown as % of baseline. Mean ± SEM is plotted. (D, E) ECAR at 2 h post-treatment of (D) MuTuDCs treated as in (C) or (E) RAW 264.7 cells ectopically expressing indicated receptors or transduced with EV stimulated ± Zym-D. $N = 3$–6 per group. Data normalised as in (C). Mean ± SEM is plotted. (F) Schematic of major metabolic pathways downstream of glucose catabolism. PEP phosphoenolpyruvate, PKM2 pyruvate kinase M2, S7P sedoheptulose 7-phosphate, 3-PG 3-phosphoglycerate, OAA oxaloacetate, α-KG α-ketoglutarate, TCA tricarboxylic acid, PPP pentose phosphate pathway. (G) Heatmap showing metabolites in untreated KO/C9 MuTuDCs or KO/C9, KO/C9(I6G), or KO/C7::C9 MuTuDCs treated with 12.5 nM DNGR-1L (log2 fold change of levels at 60 min compared to 0 min post-stimulation). F6P fructose 6-phosphate, F1,6BP fructose 1,6-bisphosphate, DHAP dihydroxyacetone phosphate, SAM S-adenosyl methionine, 2-HG 2-hydroxyglutarate. Colour bars along the right side of the graph correspond to schematic in (F). (H) Fractional labelling of metabolites in same groups described and treated as in (G) cultured with uniformly-labelled U-$^{13}$C-glucose introduced at the time of stimulation. Mean ± SEM from five biological replicates is plotted. (I, J) Left, glycolytic capacity (maximum ECAR after rotenone and antimycin A injection) and right, ECAR and oxygen consumption rate (OCR) of MuTuDCs treated for <1 h with (I) C3K (PKM2 inhibitor) or (J) DASA-58 (PKM2 agonist). (I, J) Data are normalised to baseline measurement immediately after injection with PKM2 drugs and shown as % of baseline. Mean ± SEM is plotted ($n = 4$ per treatment and 9 for untreated samples). (K) Flow cytometric analysis of % I-A/E$^{hi}$ CD86$^{hi}$ cells of indicated MuTuDCs stimulated ± 12.5 nM DNGR-1L in the presence of DMSO (Control), 10 μM C3K, or 40 μM DASA-58. Data are normalised to control condition for each cell line. Mean ± SEM from triplicate measurements is plotted. Data are representative of two independent experiments (C–E, I–K). Data were analysed using Tukey-corrected two-way ANOVA (C–E, H) or one-way ANOVA (I–K). Significant values comparing against untreated samples are plotted (C–E, H–K). (C) ****$P < 0.0001$, (D) **$P = 0.0048$, ****$P < 0.0001$, (E) **$P = 0.0063$, ***$P = 0.0004$, ****$P < 0.0001$, (H) *$P = 0.0362$, ****$P < 0.0001$, (I) **$P = 0.0032$, ****$P < 0.0001$, (J) ***$P = 0.0005$, ****$P < 0.0001$, (K) *$P = 0.0309$, ***$P = 0.0006$, ****$P < 0.0001$. See also Fig. EV3. Source data are available online for this figure.

and analysed by mass spectrometry (Fig. 4A). As expected, SYK was one of the top three proteins found associated with the tyrosine-phosphorylated C9 and C9(I6G) cytoplasmic tails (Fig. 4B) (Henry et al, 2023a; Huysamen et al, 2008; Sancho et al, 2009). Among the other most represented proteins pulled down by the phosphorylated peptides were scaffolding adaptor GRB2 and phosphatases SHP-1, SHP-2 and SHIP1 (Fig. 4B). Notably, SHP-1 has been previously identified as a target for phosphorylation following DNGR-1 triggering in cDC1s (Del Fresno et al, 2018). Src family kinase LYN and other known proteins associated with tyrosine kinase mediated signal transduction, such as PLCγ2, CSK, and SHC1, were also associated to a greater extent with C9-7pY and C9(I6G)-7pY peptides compared to the non-phosphorylated versions (Sancho and Reis e Sousa, 2012).

Few proteins were differentially associated with the phosphorylated tails of C9 versus C9(I6G), suggesting that the I6G mutation does not markedly affect which proteins can be stably recruited to DNGR-1 upon ligand engagement. However, the glycine substitution might differentially regulate the kinetics of association or the activity of the proteins that do associate with DNGR-1. To test this, we examined signalling dynamics following DNGR-1L stimulation of C9 KO and KO/C9, KO/C9(I6G) MuTuDCs (Fig. 4C). As a positive control, we confirmed that SYK pY352 was robustly induced after DNGR-1L stimulation in both C9- and C9(I6G)-expressing cells. Recently, we reported that DNGR-1 signalling is curtailed by K63 ubiquitination of SYK by E3 ligases CBL and CBL-B (Henry et al, 2023a). We hypothesised that C9(I6G) might relieve SYK ubiquitination, leading to more sustained signalling, but we found no significant change between KO/C9(I6G) and KO/C9 MuTuDCs (Fig. 4C).

In addition to NFAT signalling, MAPK and NF-κB are major pathways induced downstream of Dectin-1 and activatory innate immune receptors (Brown et al, 2003; Goodridge et al, 2007; Gringhuis et al, 2009; Gross et al, 2006; LeibundGut-Landmann et al, 2007; Rogers et al, 2005; Underhill et al, 2005). Consistent with this and our gene enrichment data (Fig. 2I), KO/C9(I6G) MuTuDCs displayed significantly enhanced p38 pT180/pY182 phosphorylation and more degraded IκB after DNGR-1L

stimulation compared to KO/C9 cells (Fig. 4C). We next turned our attention to phosphatases as they were among the top proteins pulled-down with DNGR-1 (Fig. 4B). As reported before (Del Fresno et al, 2018), SHP-1 was found to be phosphorylated in KO/C9 MuTuDCs after DNGR-1L treatment (Fig. 4C). Induction of SHP-1 pY564 signal trended slightly higher ( < 1.5-fold) in C9(I6G) expressing cells. Strikingly, SHIP1 and SHP-2 were phosphorylated to a significantly greater extent in KO/C9 MuTuDCs compared to KO/C9(I6G) cells (Fig. 4C). Given these results, we wondered whether activation of phosphatases downstream of DNGR-1 signalling restricts its ability to activate cDC1. We generated SHP-1, SHP-2, and SHIP1 deficient KO/C9 MuTuDCs (Fig. EV4A). Surprisingly, when we stimulated our phosphatase KO cells with DNGR-1L, only SHIP1 deficiency rescued cDC1 activation (Figs. 4D and EV4B). Expression of migratory and costimulatory molecules was significantly enhanced in SHIP1 KO compared to untreated cells whereas SHP-1 KO and SHP-2 KO behaved similarly to WT cells. Pharmacological inhibition of SHIP1 phenocopied our genetic loss of function results (Fig. EV4C). Additionally, SHIP1 deficient KO/C9 MuTuDCs were able to rapidly increase glycolysis in response to DNGR-1L, consistent with their rescued activation (Fig. EV4D,E).

Phosphatases SHP-1, SHP-2, and SHIP1 are recruited via their SH2 domains to receptors that contain immunoreceptor inhibitory motifs (ITIMs) (Blank et al, 2009; Lorenz, 2009). The ITIM consensus sequence V/I/S/LxYxxI/V/L bears resemblance to the hemITAM sequence of DNGR-1 (IYTSL). In addition to being more strongly phosphorylated (Fig. 4C), we wondered whether SHIP1 might be recruited to a greater extent to C9 versus C9(I6G) signalling complexes. When we incubated MuTuDCs with α-DNGR-1 IgG-coupled beads, we observed by confocal microscopy that both the mean fluorescence intensity of SHIP1 pY1021 associated with individual phagosomes and the proportion of SHIP1 pY1021$^{+}$ phagosomes was significantly greater in KO/C9 than in KO/C9(I6G) MuTuDCs (Fig. 4E). Taken together these data indicate that DNGR-1 signalling recruits and activates SHIP1, which in turn limits the ability of the receptor to activate cDC1.

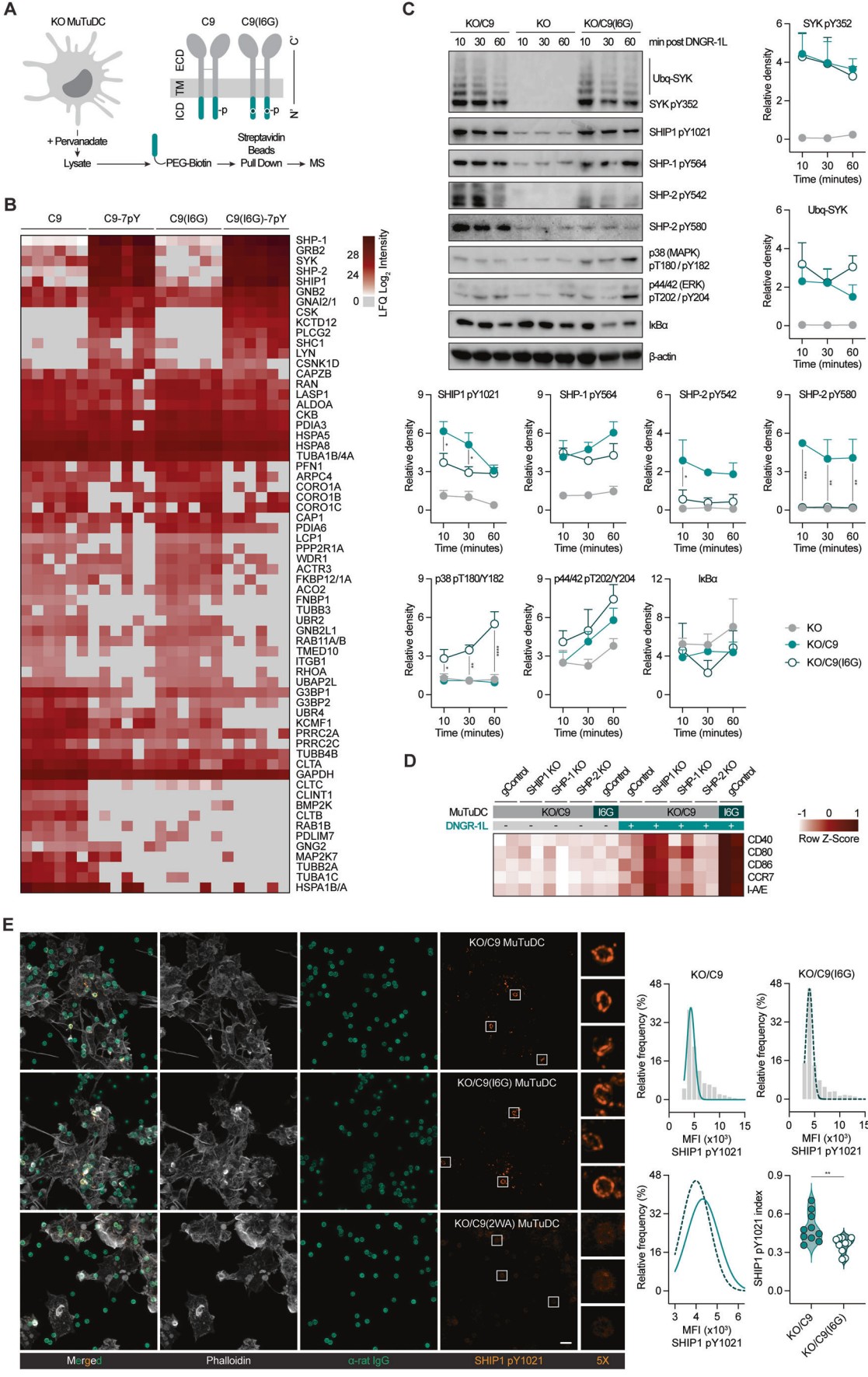

**Figure 4.  SHIP1 tempers cDC1 activation by DNGR-1.**

(A) Schematic of pull-down performed against the intracellular cytoplasmic domain (ICD) of C9, C9(I6G), or C9 and C9(I6G) 7Y-phosphorylated peptides incubated with lysates from C9 KO MuTuDCs. (B) Heatmap of label-free quantification (LFQ) intensities from samples outlined in (A) analysed by mass spectrometry. (C) Western blot analysis of C9 KO MuTuDCs or those reconstituted with C9 or C9(I6G) treated with 12.5 nM DNGR-1L. Representative images (left) and densitometry analyses of indicated proteins normalised to β-actin signal (x10). Densities represent ≥ three independent experiments shown as mean ± SEM. Ubq-SYK ubiquitinated-SYK pY352. (D) Heatmap of surface protein MFIs assessed by flow cytometry in indicated MuTuDC lines edited with control guides (gControl) or guides targeting SHIP1, SHP-1, or SHP-2 cultured overnight ± 12.5 nM DNGR-1L. Data shown are relative to untreated samples. (E) Confocal microscopic analysis of SHIP1 pY1021 in KO/C9, KO/C9(I6G), and KO/C9(2WA) MuTuDCs co-cultured with α-DNGR-1 IgG coupled beads. Representative images (left). SHIP1 pY1021$^+$ phagosomal MFI were binned and plotted against their relative distribution with a fitted curve (top right). Violin plot quantification of SHIP1 pY1021$^+$ phagosomes (bottom right) of internalised beads (SHIP1 pY1021 index). Data are representative of two independent experiments (D, E). Data were analysed using Tukey-corrected two-way ANOVA (C) or unpaired $t$ test (E). Only significant values observed between KO/C9 and KO/C9(I6G) treated samples plotted (C, E). (C) SHIP1 pY1021 10 min *$P$ = 0187, 30 min *$P$ = 0.0390; SHP-2 pY542 *$P$ = 0.0338; SHP-2 pY580 10 min ***$P$ = 0.0002, 30 min **$P$ = 0.0038, 60 min **$P$ = 0.0028; p38 pT180/Y182 10 min *$P$ = 0.0475, **$P$ = 0.0064, 60 min ****$P$ < 0.0001, (E) **$P$ = 0.0042. See also Fig. EV4. Source data are available online for this figure.

## An activatory mutant DNGR-1 displays compromised cross-presentation activity

Conversion of the isoleucine adjacent to the tyrosine in the hemITAM to glycine rescues activation by DNGR-1 partly by relieving negative regulation by SHIP1. We next examined the impact of the glycine substitution on DNGR-1-dependent cross-presentation of dead cell-associated antigens. Triggering of OVA-specific CD8$^+$ OT-I cells after co-culture with MuTuDCs incubated with DNGR-1L-OVA-coupled beads or dead cell-associated OVA was less efficient in KO/C9(I6G) and KO/C7::C9 MuTuDCs compared to KO/C9-expressing cells (Fig. 5A,B). However, all genotypes were equally effective at presenting exogenous peptide and internalising DNGR-1L-bearing beads (Fig. 5C–E). MuTuDCs lacking DNGR-1 (KO) or expressing 2WA mutant (KO/C9(2WA)) served as controls in these experiments (Fig. 5A–E). We obtained similar results using an alternative dead cell-associated antigen, influenza virus nucleocapsid protein (NP), in cross-presentation assays with NP-specific CD8$^+$ F5 T cells (Fig. EV5A,B). Thus, converting DNGR-1 into an activating receptor impairs its ability to signal to induce cross-presentation.

To determine the impact of the I6G substitution on phagosome-to-cytosol translocation downstream of DNGR-1, we assessed the recruitment of lysenin-mCherry to phagosomes of MuTuDCs that had internalised α-DNGR-1 IgG-coupled beads (Figs. 5F–H and EV5C). Again, particle uptake was identical between KO/C9 and KO/C9(I6G) MuTuDCs (Fig. 5F). However, KO/C9(I6G)-expressing cells accumulated significantly fewer lysenin-mCherry$^+$ phagosomes than KO/C9 MuTuDCs (Figs. 5G,H and EV5C), consistent with their compromised cross-presentation of dead cell- and ligand-associated material (Fig. 5A,B). Finally, we assessed whether genetic ablation of SHIP1 similarly reduced the ability of MuTuDCs to cross-present antigens via the DNGR-1-dependent pathway. Notably, although peptide presentation and DNGR-1L bead uptake were unaffected, SHIP1-deficient MuTuDCs displayed an impairment in cross-presentation of DNGR-1L-OVA-coupled beads or dead cell-associated OVA to OT-I T cells, albeit less pronounced than the one observed in cells expressing C9(I6G) (Fig. 5I–L). Overall, these data suggest that introduction of a glycine next to the hemITAM of DNGR-1 or loss of SHIP1 renders the receptor less efficient at mediating phagosomal damage and limits cross-presentation of ligand-associated antigens.

## Efficient cross-presentation is balanced against activation

We hypothesised that there might be a trade-off between the ability of CLRs such as DNGR-1 and Dectin-1 to mediate cDC1 activation versus cross-presentation. In this scenario, introducing the reverse mutation into Dectin-1 (G14I; see Fig. 2A) to make it "DNGR-1-like" would impair its capacity to promote activation and concomitantly boost its ability to facilitate cross-presentation. We first assessed the capacity of C7(G14I) to signal for myeloid cell activation. RAW 264.7 cells ectopically expressing C7 and C9(I6G)::C7 robustly upregulated markers of activation and released TNF-α in response to Zym-D (Figs. 6A and EV6A). In contrast, cells expressing the C7(G14I) mutant, like those expressing C9::C7, did not become activated in response to Zym-D (Fig. 6A) despite all cells expressing similar levels of the receptors (Fig. EV6B).

We next tested the effect of the C7(G14I) cytoplasmic tail mutation on activation and cross-presentation by MuTuDCs. Both C7(G14I)::C9 and C7::C9 chimeric receptors were equally expressed at the cell surface (Fig. EV6C), suggesting that there was no inherent defect in the ability of the C7(G14I)::C9 chimera to properly fold and traffic within the cell. Furthermore, consistent with the RAW 264.7 data, KO/C9(I6G) and KO/C7::C9 MuTuDCs upregulated costimulatory molecules, CCR7 and secreted chemokines after DNGR-1L stimulation, while MuTuDCs expressing C7(G14I)::C9 or C9 exhibited little change compared to untreated controls (Figs. 6B and EV6D). However, the C7(G14I)::C9 chimeric receptor displayed an impairment in mediating phagocytosis of DNGR-1L-coupled beads compared to the C7::C9 chimera (Fig. 6C), which made it difficult to assess its activity in cross-presentation. To remedy this issue, we isolated by FACS MuTuDCs that had internalised a single DNGR-1L-OVA bead (Fig. 6D) (Canton et al, 2021; Schnorrer et al, 2006), thereby normalising for antigen load before using the cells in a cross-presentation assay. Consistent with our hypothesis, we observed greater IFN-γ released from OT-I cells co-cultured with KO/C7(G14I)::C9 single bead$^+$ MuTuDCs compared to single bead$^+$ KO/C7::C9 cells (Fig. 6E). Both cell lines were equally able to stimulate OT-I when pulsed with OVA peptide (Fig. 6E). Altogether our data suggest that making the hemITAM of Dectin-1 more "DNGR-1-like" results in a gain-of-function in cross-presentation and a loss-of-function in cell activation.

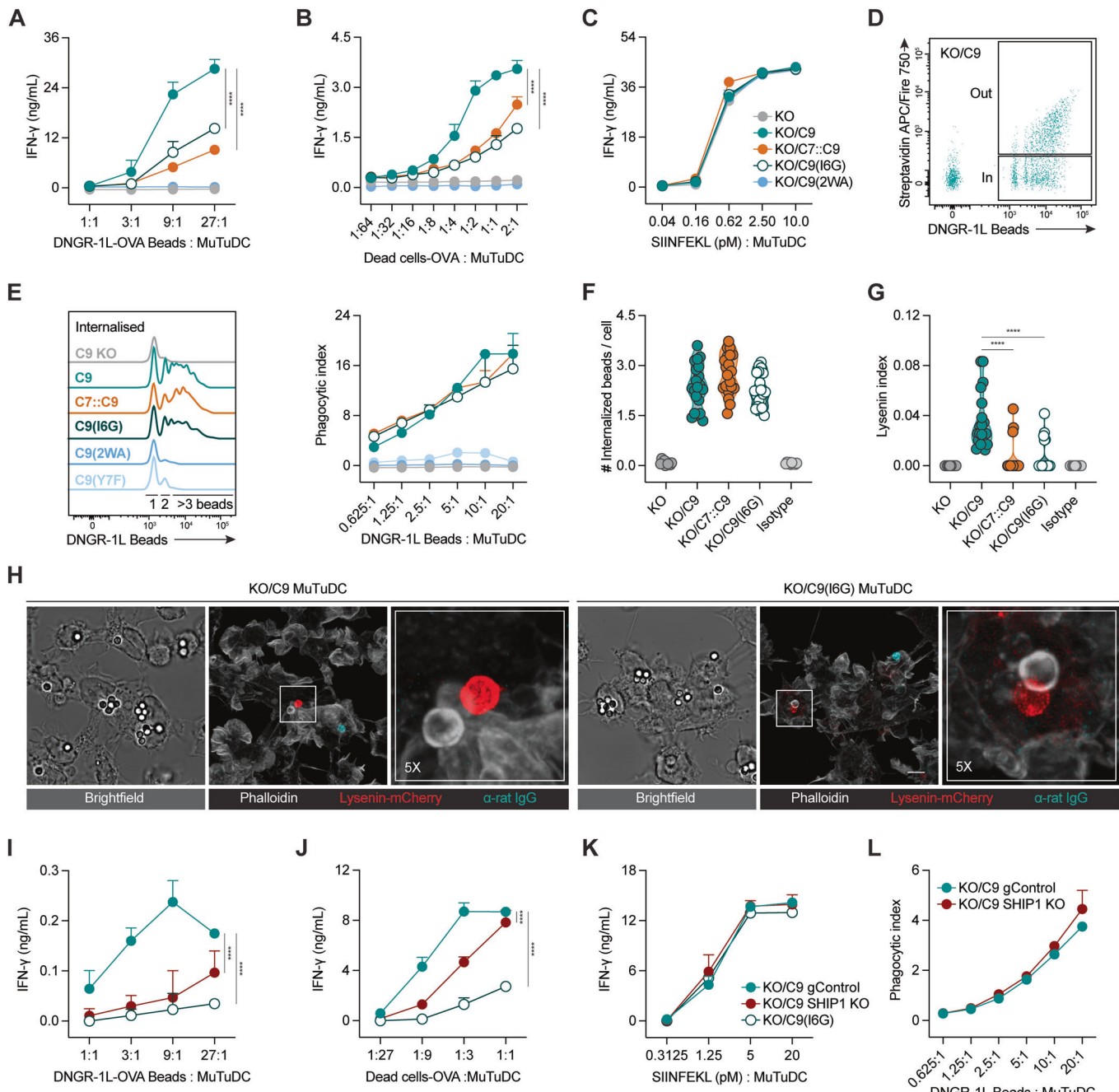

# Discussion

cDCs express an array of receptors that detect signs of infection and damage, as well as facilitate the capture and processing of antigens for presentation to T cells (Cabeza-Cabrerizo et al, 2021). One such receptor, DNGR-1, is expressed by cDC1s and signals for phagosomal rupture and cross-presentation upon binding to F-actin exposed by cell debris after necrophagy (Ahrens et al, 2012; Canton et al, 2021; Sancho et al, 2009; Schulz et al, 2018). Innate immune receptors, including related CLRs such as Dectin-1, often signal to promote proinflammatory gene expression and other features associated with myeloid cell activation. Notably, we

demonstrate that DNGR-1 is fundamentally distinct from such innate immune receptors in that its signalling recruits the SHIP1 phosphatase and does not activate cDC1s. We reveal that replacement of a single residue adjacent to the tyrosine in the DNGR-1 hemITAM motif or loss of SHIP1 rescues the capacity of the receptor to activate cDC1s but compromises its ability to promote cross-presentation. Interestingly, the converse is seen when the reverse mutation is introduced into Dectin-1. These findings uncover a trade-off between the role of DNGR-1 in cross-presentation and cellular activation, suggesting that the two processes are fundamentally incompatible or, more likely, that the recognition of cell death is subject to multiple checkpoints in

**Figure 5. An activatory DNGR-1 exhibits compromised ability to induce cross-presentation.**

(A–C) ELISA for IFN-γ release from OT-I T$_E$ cells co-cultured with C9 KO MuTuDCs or those reconstituted with indicated receptors incubated with (A) DNGR-1L-OVA-coupled beads, (B) OVA-dead cells, or (C) SIINFEKL peptide. Mean ± SD from biological (B, C) duplicates or (A) triplicates are plotted. (D, E) Uptake of DNGR-1L-coupled beads by C9 KO MuTuDCs or those reconstituted with indicated receptors assessed by flow cytometry. (D) Representative plot depicting the use of post-uptake streptavidin staining to distinguish MuTuDCs that have internalised (in) biotinylated-DNGR-1L coupled beads from those attached to surface DNGR-1 (out). (E) Representative histogram of cells treated 20:1 with beads (left) and phagocytic index (% internalised beads × bead MFI/arbitrary unit) mean ± SD from biological duplicates (right). (F–H) Confocal microscopic analysis of lysenin-mCherry fusion protein-expressing C9 KO MuTuDCs or those reconstituted with indicated receptors co-cultured with α-DNGR-1 IgG-coupled or isotype IgG-coupled beads. Beads not internalised marked by α-rat IgG staining. (F) Violin plot quantification of internalised beads per cell per field of view (n = 20). (G) Violin plot of lysenin-mCherry$^+$ phagosomes per cells in field of view (Lysenin index). (H) Representative images of lysenin-mCherry$^+$ phagosomes from indicated MuTuDCs, scale bar = 5 µm. (I–K) ELISA for IFN-γ release from OT-I T$_E$ cells co-cultured with KO/C9(I6G) or SHIP1 sufficient (guide control; gControl) or deficient KO/C9 MuTuDCs incubated with (I) DNGR-1L-OVA-coupled beads, (J) OVA-dead cells, or (K) SIINFEKL peptide. Mean ± SD from (I–K) biological duplicates or quadruplets (KO/C9 SHIP1 KO) are plotted. (L) Uptake of DNGR-1L-coupled beads by KO/C9 SHIP1 sufficient or deficient MuTuDCs assessed by flow cytometry. Phagocytic index plotted with mean ± SD from biological triplicates. Data are representative of two (F–L) or ≥ three (A–C) independent experiments. Data were analysed using Tukey-corrected two-way ANOVA with only significant values observed between KO/C9 and KO/C9(I6G) or KO/C7::C9 MuTuDCs plotted in (A–C, E–G) or between KO/C9 gControl and KO/C9(I6G) or KO/C9 SHIP1 KO MuTuDCs in (I–K). (A, B, G, I, J) ****$P < 0.0001$. See also Fig. EV5. Source data are available online for this figure.

order to prevent auto-inflammatory responses that might ensue from (mis)triggering of a single dead cell receptor.

SHIP1, a 5' inositol phosphatase, has been implicated in regulating antigen presentation in myeloid cells, particularly in cDCs, and SHIP1 deficiency is known to affect phagosomal processing and immune activation (Gold et al, 2016; Kamen et al, 2008; Kamen et al, 2007). Indeed, we observed an impairment in DNGR-1 dependent cross-presentation in SHIP1 deficient cells. SHIP1 likely modulates the phosphoinositide signalling landscape within phagosomes (Kamen et al, 2008; Kamen et al, 2007), affecting the ability of cDC1s to capture, process, and present antigens. In addition, SHIP1 may control phagosomal rupture, a critical step for antigen release and cross-presentation (Blanco-Menendez et al, 2015; Canton et al, 2021). Whether signals associated with activation might be incompatible with the ability of SHIP1 to mediate these functions remains unclear. Intriguingly, SHIP1 deficiency rescued glycolytic responses to DNGR-1 ligand engagement and it will be interesting to explore whether a specific type of metabolic state is optimal for cross-presentation versus activation. Further research into the interaction between SHIP1 and DNGR-1 signalling could reveal insights into how these pathways integrate to regulate immune responses.

Unlike isoleucine, a branched-chain amino acid, glycine possesses a single hydrogen atom in its side chain. Its small size confers additional flexibility in protein structure (Yan and Sun, 1997), which may be helpful to receptor function. In the context of ion channels, for example, glycine can allow for slight conformational changes or "hinge" movements that are necessary for the opening and closing of the receptor (Ding et al, 2005). The ability of glycine to form tight turns or bends in the polypeptide chain might help position how receptors congregate in the membrane or recruit other proteins, which can be critical to signalling dynamics (Chou and Fasman, 1977). Indeed, mutation of the tyrosine-adjacent glycine in Dectin-1 with isoleucine was enough to abolish myeloid activation in multiple cell types. Previous work also showed that substitution of the homologous glycine residue present within CLEC-2 with alanine, which is comparable in size but contains a bulkier side chain, also impairs its ability to signal effectively (Fuller et al, 2007). Interestingly, the positioning of both isoleucine and glycine is conserved in DNGR-1 and Dectin-1, respectively, across all the species we analysed, which suggests that our results are likely

to be applicable to human cells. Furthermore, the glycine position is also conserved in other activating hemITAM-bearing receptors (Bauer and Steinle, 2017; Zelenay et al, 2012). This suggests that DNGR-1 may have diverged from other hemITAM-bearing CLRs and acquired the ability to facilitate cross-presentation of dead cell-associated antigens while losing activatory capacity. As such, DNGR-1 represents a class of innate immune receptor that selectively helps cDCs retrieve antigenic information from dead cell debris.

Both the recognition of ligand and downstream signalling by DNGR-1 are subject to stringent negative regulation (Henry et al, 2023b). For example, F-actin is normally sequestered intracellularly and not exposed during apoptotic cell death (Ahrens et al, 2012; Sancho et al, 2009; Zhang et al, 2012). Furthermore, DNGR-1 engagement of F-actin is also inhibited by secreted gelsolin, which is present at high concentrations in extracellular fluids (Giampazolias et al, 2021). Additionally, SYK is rapidly ubiquitinated by CBL and CBL-B after DNGR-1 signalling, which restrains DNGR-1-dependent phagosomal damage (Henry et al, 2023a). Finally, DNGR-1 expression is highly restricted to cDC1s, a rare immune cell subset within tissues (Cabeza-Cabrerizo et al, 2021). These various layers of regulation, in addition to the findings reported here, lead to the viewpoint that DNGR-1 is carefully controlled to avoid pathological immune activation. It is possible that cDC1 activation by a single dead cell-recognising receptor would pose an evolutionarily unacceptable risk for inducing autoimmune responses against self-antigens. Restraining DNGR-1 activity may ensure that cDC1 activation leading to cross-priming occurs only in the presence of PAMPs (e.g., double stranded RNA within corpses of virally infected cells (Schulz et al, 2005)) or upon coincident detection of additional DAMPs such as may be present during pathology (e.g., cancer (Giampazolias et al, 2021)), and not under normal, non-pathological conditions. By selectively controlling the ability of cDC1s to "read-out" the antigenicity of dead cells and separating it from DAMP-induced cDC1 activation, DNGR-1 may also participate in central tolerance, selecting against developing T cells with receptors that possess high affinity for self (dead cell-associated) antigens (Nurieva et al, 2011; Perry and Hsieh, 2016; Perry et al, 2018; Villadangos and Schnorrer, 2007). Similarly, a separation between signalling for antigen uptake and cell activation has also been

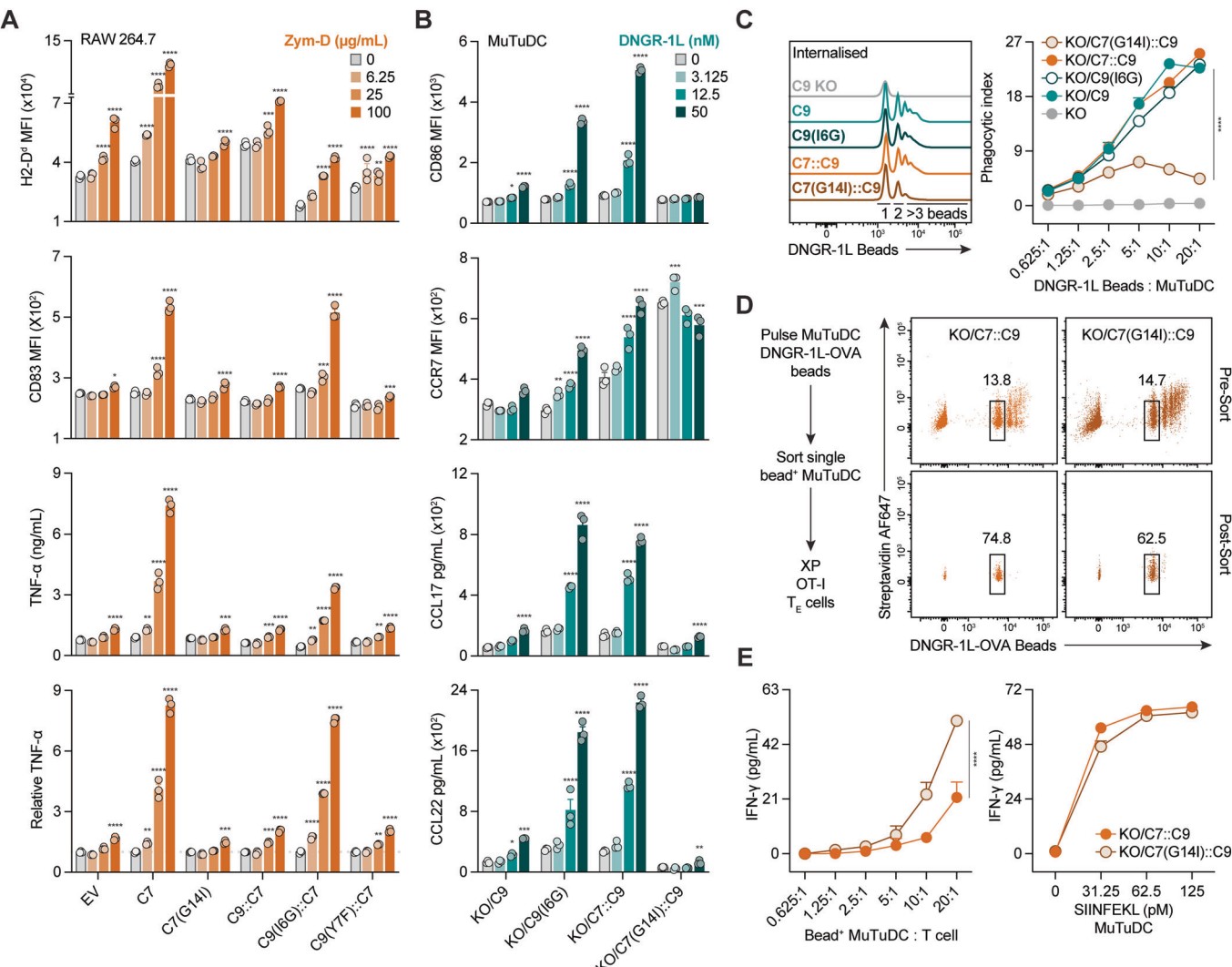

**Figure 6. A "DNGR-1-like" Dectin-1 receptor displays reduced ability to induce activation but gains cross-presentation capacity.**

(A, B) Flow cytometric analyses of surface proteins and ELISA of indicated proteins released in culture supernatants from (A) RAW 264.7 cells ectopically expressing indicated receptors or transduced with EV stimulated overnight ± Zym-D or (B) C9 KO MuTuDCs reconstituted with indicated receptors stimulated overnight ± DNGR-1L. Data shown as mean ± SEM from biological triplicates. Note that the TNF-α data in (A) are plotted in two different ways (absolute versus relative concentration to untreated controls) to emphasise the similarity between C7 and C9(I6G)::C7. (C) Uptake of DNGR-1L-coupled beads by C9 KO MuTuDCs or those reconstituted with indicated receptors assessed by flow cytometry. Plotted as phagocytic index (% internalised beads x bead MFI of bead⁺ cells/arbitrary unit) mean ± SD from biological duplicates. (D, E) C9 KO MuTuDCs reconstituted with C7::C9 or C7(G14I)::C9 receptors co-cultured with DNGR-1L-OVA coupled beads. Single internalised bead⁺ cells were subsequently sorted and co-cultured with OT-I $T_E$ cells. (D) Schematic of experiment (left) and representative flow plots from pre- and post-sort enrichment (right). (E) ELISA for IFN-γ released from OT-I $T_E$ cells co-cultured with MuTuDCs from (D) or MuTuDCs loaded with exogenous SIINFEKL peptide. Mean ± SD from biological duplicates is plotted. Data are representative of two (D, E) or ≥ three (A–C) independent experiments. Data were analysed using Tukey-corrected two-way ANOVA (A–C, E). Significant values comparing against untreated controls (A, B) or between KO/C7(G14I)::C9 (E) are plotted. (A) H2-D$^d$ **$P = 0.0043$, ***$P = 0.0006$, ****$P < 0.0001$; CD83 *$P = 0.0374$, ***$P = 0.0002$, ****$P < 0.0001$; TNF-α **$P = 0.0017$ (C7), $P = 0.0067$ (C9(Y7F)::C7, ***$P = 0.0004$ (C7(G14I)), $P = 0.0006$ (C9::C7), ****$P < 0.0001$, (B) CD86 *$P = 0.0469$, ****$P < 0.0001$; CCR7 **$P = 0.0097$, ***$P = 0.0004$, ****$P < 0.0001$; CCL17 ****$P < 0.0001$; CCL22 *$P = 0.0217$, **$P = 0.0018$, ****$P < 0.0001$, (C, E) ****$P < 0.0001$. See also Fig. EV6. Source data are available online for this figure.

demonstrated for other CLRs and FcRs (Dudziak et al, 2007; Hawiger et al, 2001; Lehmann et al, 2017). Finally, even if it does not signal for activation, is possible that DNGR-1 may nevertheless help indirectly to decode dead cell adjuvanticity. For example, DNGR-1-dependent phagosomal disruption may facilitate the transfer of DAMPs such as nucleic acids to the cytosol of cDC1s, which then engage cytosolic innate immune receptors (Henry et al, 2023b; Woo et al, 2014).

In summary, our study highlights the complex regulatory mechanisms that govern DNGR-1 signalling in cDC1s, particularly the interplay between immune activation and cross-presentation. The physiological consequences of perturbing this balancing act from DNGR-1 remains to be fully explored and assessed in vivo. Yet the insights gained from this study may have implications for designing strategies to modulate cDC activity in cancer immunotherapy and autoimmunity.

# Methods

## Reagents and tools table

| Reagent/resource | Reference or source | Identifier or catalog number |
|---|---|---|
| **Experimental models** | | |
| Mouse: C57BL/6 J | Francis Crick Institute | RRID:IMSR_JAX:000664 |
| Mouse: Clec9a^Cre/Cre | Francis Crick Institute | Allele MGI ID: 5502446 |
| Mouse: OT-I RAG1 KO | Francis Crick Institute | Allele MGI ID: 3054907, 2448994 |
| Mouse: F5 RAG1 KO | Francis Crick Institute | Allele MGI ID: 3706698, 1857241 |
| Cell line: MuTuDC1940 | Hans Acha-Orbea Lab | |
| Cell line: B3Z | Nilabh Shastri Lab | |
| Cell line: BRAFV600E 5555 melanoma | Francis Crick Institute | |
| Cell line: RAW264.7 | Francis Crick Institute | |
| Cell line: GP2-293 | Francis Crick Institute | |
| Cell line: Phoenix-ECO | Francis Crick Institute | |
| Cell line: HeLa | Francis Crick Institute | |
| **Recombinant DNA** | | |
| pMD2.G | Addgene | Cat. #12259 |
| pFB-Empty Vector-IRES-GFP | Caetano Reis e Sousa Lab | N/A |
| pFB-C9-IRES-GFP | Caetano Reis e Sousa Lab | N/A |
| pFB-C9(I6G)-IRES-GFP | Caetano Reis e Sousa Lab | N/A |
| pFB-C9(2WA)-IRES-GFP | Caetano Reis e Sousa Lab | N/A |
| pFB-C9(Y7F)-IRES-GFP | Caetano Reis e Sousa Lab | N/A |
| pFB-C7::C9-IRES-GFP | Caetano Reis e Sousa Lab | N/A |
| pFB-C7(G14I)::C9-IRES-GFP | Caetano Reis e Sousa Lab | N/A |
| pFB-C7-IRES-GFP | Caetano Reis e Sousa Lab | N/A |
| pFB-C7(G14I)-IRES-GFP | Caetano Reis e Sousa Lab | N/A |
| **Antibodies** | | |
| CD16/32 (Fc block) | BD Pharmingen | Clone 2.4G2 |
| CD11c | Biolegend | Clone N418 |
| CD40 | Biolegend | Clone 3/23 |
| CD45R/B220 | Biolegend | Clone RA3-6B2 |
| CD80 | Biolegend | Clone 16-10A1 |
| CD83 | BD Biosciences | Clone Michel-19 |

| Reagent/resource | Reference or source | Identifier or catalog number |
|---|---|---|
| CD86 | Biolegend; eBioscience | Clone GL-1 |
| CD172a/Sirpα | Biolegend | Clone P84 |
| CD197/CCR7 | BD Biosciences | Clone 4B12 |
| DNGR-1/CLEC9A | Francis Crick Institute; Biolegend | Clone 1F6 |
| DNGR-1/CLEC9A | Francis Crick Institute | Clone 7H11 |
| Dectin-1/CLEC7A | Bio-Rad | Clone 2A11 |
| H2-K^b | Biolegend | AF6-88.5 |
| H2-D^d | BD Pharmingen | Clone 34-2-12 |
| H2-K^d/D^d | Biolegend | Clone 34-1-2S |
| I-A/E | Thermo Fisher Scientific | Clone M5/114.15.2 |
| Rat IgG isotype control | Francis Crick Institute | Clone MAC49 |
| XCR1 | Biolegend | Clone ZET |
| IL-12 p40 | Biolegend | Clone C15.6 |
| IFN-γ | BD Biosciences | Clone R4-6A2, XMG1.2 |
| IκBα | Cell Science Technologies | Clone 44D4 |
| p38 pT180/pY182 | Cell Science Technologies | Clone D3F9 |
| p44 pT202/pY204 | Cell Science Technologies | Clone 9101 |
| SHP-1 pY564 | Cell Science Technologies | Clone D11G5 |
| SHP-2 pY542 | Cell Science Technologies | Clone 3751 |
| SHP-2 pY580 | Cell Science Technologies | Clone 3703 |
| SYK pY352 | Cell Science Technologies | Clone 65E4 |
| SHIP1 pY1021 | STEMCELL Technologies | Cat. #60142 |
| β-actin-HRP | Cell Science Technologies | Clone 13E5 |
| α-mouse IgG-HRP | Cell Science Technologies | Cat. #7076 |
| α-rabbit IgG-HRP | Cell Science Technologies | Cat. #7074 |
| Donkey α-rabbit IgG AF647 Plus | Fisher Scientific | Cat. #A32795 |
| **Oligonucleotides and other sequence-based reagents** | | |
| Taqman Ccl17 | Thermo Fisher Scientific | Assay ID: Mm00516136_m1 |
| Taqman Ccl22 | Thermo Fisher Scientific | Assay ID: Mm00436438_m1 |
| Taqman Tnf | Thermo Fisher Scientific | Assay ID: Mm00443258_m1 |
| Taqman Actb | Thermo Fisher Scientific | Assay ID: Mm00607939_s1 |

| Reagent/resource | Reference or source | Identifier or catalog number |
|---|---|---|
| IDT Alt-R CRISPR-Cas9 *Inpp5d* sgRNA #1 | IDT | Cat. #Mm.Cas9.INPP5D.1.AA |
| IDT Alt-R CRISPR-Cas9 *Inpp5d* sgRNA #2 | IDT | Cat. #Mm.Cas9.INPP5D.1.AB |
| IDT Alt-R CRISPR-Cas9 *Ptpn6* sgRNA #1 | IDT | Cat. #Mm.Cas9.PTPN6.1.AA |
| IDT Alt-R CRISPR-Cas9 *Ptpn6* sgRNA #2 | IDT | Cat. #Mm.Cas9.PTPN6.1.AB |
| IDT Alt-R CRISPR-Cas9 *Ptpn11* sgRNA #1 | IDT | Cat. #Mm.Cas9.PTPN11.1.AA |
| IDT Alt-R CRISPR-Cas9 *Ptpn11* sgRNA #2 | IDT | Cat. #Mm.Cas9.PTPN11.1.AB |
| **Chemicals, enzymes and other reagents** | | |
| DMEM | Gibco | Cat. #41966029 |
| RPMI 1640 | Gibco | Cat. #31870025 |
| IMDM | Gibco | Cat. #12440053 |
| Fetal calf serum | Sigma-Aldrich | Cat. #F7524 |
| L-Glutamine | Gibco | Cat. #25030081 |
| Penicillin-streptomycin | Gibco | Cat. #15070063 |
| HEPES | Gibco | Cat. #15630080 |
| MEM non-essential amino acids solution | Gibco | Cat. #11140050 |
| Sodium pyruvate | Gibco | Cat. #11360070 |
| 2-mercaptoethanol | Gibco | Cat. #31350010 |
| FLT3L | R&D Systems | Cat. #427-FL |
| RBC lysis buffer | Thermo Fisher Scientific | Cat. #15474569 |
| LS columns | Miltenyi Biotec | Cat. #130-042-401 |
| SIINFEKL peptide | Francis Crick Institute | |
| ASNENDAM peptide | Francis Crick Institute | |
| Recombinant IL-2 | Peprotech | Cat. #212-12-20UG |
| GeneJuice | Novagen | Cat. #70967 |
| Opti-MEM | Gibco | Cat. #31985070 |
| Polybrene | Sigma-Aldrich | Cat. #TR-1003-G |
| LIVE/DEAD fixable blue | Thermo Fisher Scientific | Cat. #L23105 |
| LIVE/DEAD fixable aqua | Thermo Fisher Scientific | Cat. #L34957 |
| DPBS | Gibco | Cat. #14190144 |
| EDTA | Thermo Fisher Scientific | Cat. #15575-020 |
| GolgiPlug | BD Biosciences | Cat. #555029 |
| CytoFix/CytoPerm Kit | BD Biosciences | Cat. #554714 |
| LEGENDplex CBA Kit | Biolegend | Cat. #740846 |
| Actin | Cytoskeleton | Cat. #AKL99 |
| Myosin II | Cytoskeleton | Cat. #MY02-A |
| Biotinylated actin | Cytoskeleton | Cat. #AB07-A |

| Reagent/resource | Reference or source | Identifier or catalog number |
|---|---|---|
| General actin buffer (G-buffer) | Cytoskeleton | Cat. #BSA01 |
| Actin polymerization buffer (F-buffer) | Cytoskeleton | Cat. #BSA02 |
| Streptavidin coated beads | Polysciences | Cat. #24160-5 |
| FluoSpheres NeutrAvidin-labeled red microspheres | Life Technologies | Cat. #F8775 |
| Streptavidin Fluoresbrite YG microspheres | Polysciences | Cat. #24159-1 |
| Ovalbumin | Sigma-Aldrich | Cat. #A5378 |
| BSA | Merck Life Science | Cat. #A9576 |
| DSB-X-biotinylation kit | Thermo Fisher Scientific | Cat. #D20655 |
| Hot alkali treated zymosan | Invivogen | Cat.#tlrl-zyd |
| Poly(I:C) | Invivogen | Cat. #tlrl-pic |
| Recombinant IFN-αA | R&D Systems | Cat. #10150-IF-050 |
| C3K | Universal Biologicals | Cat. #S8616 |
| DASA-58 | Merck Life Science | Cat. #SML2853 |
| 3AC | MedChemExpress | Cat. #HY-19776 |
| RMC-4550 | MedChemExpress | Cat. #HY-116009 |
| CCL17/TARC DuoSet ELISA kit | R&D Systems | Cat. #DY529 |
| CCL22/MDC DuoSet ELISA kit | R&D Systems | Cat. #DY439 |
| IL-12p40 DuoSet ELISA kit | R&D Systems | Cat. #DY2398 |
| Chlorophenolred-β-D-galactopyranoside (CPRG) | Roche | Cat. #07930097103 |
| Nunc MaxiSorp ELISA plates | Fisher Scientific | Cat. #442404 |
| Recombinant IFN-γ | Peprotech | Cat. #315-05 |
| ExtrAvidin-alkaline phosphatase | Merk Life Science | Cat. #E2636-5ML |
| SIGMAFAST p-nitrophenyl substrate | Sigma-Aldrich | Cat. #N2770 |
| CellTracker deep red dye | Thermo Fisher Scientific | Cat. #C34565 |
| Cytochalasin D | Sigma-Aldrich | Cat. #C2618-200UL |
| Alt-R S.p. Cas9 nuclease V3 | IDT | Cat. #1081059 |
| Alt-R S.p. Cas9 electroporation enhancer | IDT | Cat. #1075916 |
| Amaxa P3 Primary Cell 4D Nucleofector X kit L | Lonza | Cat. #V4XP-3024 |
| Glass coverslips | VWR | Cat. #630-1845 |
| Alexa Fluor Plus 405 Phalloidin | Thermo Fisher Scientific | Cat. #A30104 |
| Glass slides | Sail Brand | Cat. #7101 |
| Prolong Diamond Antifade Mountant | Thermo Fisher Scientific | Cat. #P36961 |
| RPMI 1640 powder no glucose or sodium bicarbonate | Sigma-Aldrich | Cat. #R1383 |

| Reagent/resource | Reference or source | Identifier or catalog number |
| --- | --- | --- |
| Poly-D-Lysine | Sigma-Aldrich | Cat. #P6407-5MG |
| Oligomycin | Sigma-Aldrich | Cat. #O4876-5MG |
| FCCP | Sigma-Aldrich | Cat. #C2920-10MG |
| Rotenone | Sigma-Aldrich | Cat. #R8875-1G |
| Antimycin A | Sigma-Aldrich | Cat. #A8675-25MG |
| DMSO | Sigma-Aldrich | Cat. #D8418-50ML |
| RPMI 1640, no glucose | Life Technologies | Cat. #11879020 |
| $^{13}$C D-glucose | Cambridge Isotope Laboratories | Cat. #CLM-1396-PK |
| LC/MS grade methanol | Fisher Scientific | Cat. #A456-212 |
| Chloroform HPLC grade | Fisher Scientific | Cat. #11390268 |
| Nuclease-free water | Thermo Fisher Scientific | Cat. #4387936 |
| LC-MS vial | Agilent | Cat. #5182-0716 |
| Laemmli sample buffer | Bio-Rad | Cat. #1610737 |
| HALT Protease and Phosphatase Inhibitor Cocktail | Thermo Fisher Scientific | Cat. #78441 |
| 4-20% Mini Protean TGX Precast Gels | Bio-Rad | Cat. #4561096 |
| QIAshredders | Qiagen | Cat. #79656 |
| RNeasy Mini kit | Qiagen | Cat. #74106 |
| Random primers | Fisher Scientific | Cat. #48190-011 |
| Superscript II reverse transcriptase | Thermo Fisher Scientific | Cat. #18064071 |
| Taqman Universal PCR master mix | Thermo Fisher Scientific | Cat. #4304437 |
| NEBNext Ultra II Directional PolyA mRNA kit | NEB | Cat. #E7760 |
| Sodium orthovanadate | Sigma-Aldrich | Cat. #450243-10 G |
| Hydrogen peroxidase | Sigma-Aldrich | Cat. #H1009-5ML |
| Cell lysis buffer | Cell Science Technologies | Cat. #9803S |
| PMSF | Sigma-Aldrich | Cat. #10837091001 |
| Pierce streptavidin magnetic beads | Thermo Fisher Scientific | Cat. #88817 |
| Brij-58 | Merck Life Science | Cat. #P5884-100G |
| **Software** | | |
| Excel | Microsoft | Version 16.100.1 |
| FlowJo | Tree Star Inc. | Version 10 |
| Prism | GraphPad | Version 10 |
| FIJI | NIH | Version 1.0 |
| ZEN Blue | Zeiss | Version 2.3 SP1 |
| ZEN Black | Zeiss | Version 2.3 |
| Xcalibur | Thermo Fisher Scientific | Version 4.2.47 |
| TraceFinder EFS | Thermo Fisher Scientific | Version 4.1 |

| Reagent/resource | Reference or source | Identifier or catalog number |
| --- | --- | --- |
| MANIC | Francis Crick Institute | Version 3.0.18 |
| RSEM package | Dewey Lab | Version 1.3.30 |
| STAR alignment algorithm | Alexander Dobin | Version 2.5.2a |
| DESeq2 package | Michael Love | Version 1.24.1 |
| R programming environment | R Core Team | Version 4.1.0 |
| MaxQuant | MaxQuant | Version 2.0.1.0 |
| Perseum Module | MaxQuant | Version 1.6.14.0 |
| **Other** | | |
| LSRFortessa | BD Biosciences | |
| LSRFortessa X-20 | BD Biosciences | |
| FACSymphony | BD Biosciences | |
| Aria III | BD Biosciences | |
| Fusion | BD Biosciences | |
| Influx | BD Biosciences | |
| Avalon | Propel Labs | |
| Spark plate reader | Tecan | |
| 4D Nucleofector | Lonza | |
| LSM880 inverted microscope | Zeiss | |
| 96-well Seahorse XF Analyzer | Agilent | |
| SpeedVac RVC 2-33 CDplus | Christ | |
| Q-Exactive Plus (Orbitrap) mass spectrometer | Thermo Fisher Scientific | |
| Vanquish UHPLC system | Thermo Fisher Scientific | |
| SeQuant Zic pHILIC column | Merck Millipore | |
| 7890B-5977A GC-MSD | Agilent | |
| DB-5MS | Agilent | |
| ImageQuant 800 | Amersham | |
| QuantStudio 3, 5, 7 RT-PCR system | Thermo Fisher Scientific | |
| HiSeq 4000 System | Illumina | |
| Ultimate U3000 HPLC | Thermo Fisher Scientific | |
| C18 Acclaim PepMap100 | Thermo Fisher Scientific | |
| EASY-Spray PepMAP RSLC | Thermo Fisher Scientific | |
| Orbitrap Eclipse Tribrid mass spectrometer | Thermo Fisher Scientific | |
| UVC Crosslinker | Hoefer | |
| Trans-Blot Turbo Transfer System | Bio-Rad | |

## Mice

C57BL/6Jax, *Clec9a*[Cre/Cre] (abbreviated C9[KI-Cre]; i.e. DNGR-1-deficient), OT-I RAG1 KO, and F5 RAG1 KO mice were bred at the

Francis Crick Institute under specific-pathogen-free conditions. All animal experiments were performed in accordance with national and institutional guidelines for animal care and were approved by the Francis Crick Institute Biological Resources Facility Strategic Oversight Committee (incorporating the Animal Welfare and Ethical Review Body) and by the Home Office, UK.

## Primary cells and cell lines

RPMI 1640 supplemented with 2 mM glutamine, 100 U/mL penicillin, 100 μg/mL streptomycin, non-essential amino acids, 10 mM HEPES, 1 mM sodium pyruvate, 50 μM 2-mercaptoethanol (all from Gibco) and 10% heat-inactivated fetal calf serum (FCS) (R10$^+$ medium) was used for all cell culture unless otherwise stated. Splenic cDC1 cell line MuTuDC1940 was a gift from Hans Acha-Orbea (Fuertes Marraco et al, 2012). MuTuDC $Clec9a^{-/-}$ sub-lines complemented with WT DNGR-1 cDNA or those harbouring W155A-W250A (2WA) or Y7F mutations were previously described (Hanc et al, 2015). B3Z cells (gift from Nilabh Shastri) containing a reporter plasmid for NFAT coupled to LacZ activity and transduced with murine DNGR-1 and SYK have been previously described (Sancho et al, 2009; Schulz et al, 2018). BRAF$^{V600E}$ 5555 melanoma were a gift from Richard Marais. RAW 264.7, GP2-293, Phoenix-ECO and HeLa cell lines were obtained from the Francis Crick Institute Cell Services Science Technology Platform (STP). Packaging cell lines GP2-293 and Phoenix-ECO were cultured in DMEM supplemented with 2 mM glutamine, 100 U/mL penicillin, 100 μg/mL streptomycin, and 10% FCS. All cell lines were authenticated for species origin and tested for mycoplasma contamination.

For cDCs grown with FLT3L (BM-FLT3L cDCs), bone marrow was extracted from hind legs and hips of mice and subjected to red blood cell (RBC) lysis (Thermo Fisher Scientific). Cells were cultured for 9 days in R10$^+$ medium with 150 ng/mL recombinant mouse FLT3L (R&D Systems). cDC1s were purified using biotinylated α-mouse XCR1 IgG (Biolegend, clone ZET), α-biotin microbeads and LS columns (both Miltenyi Biotec) according to the manufacturer's instructions. > 90% purity was checked by flow cytometry (see Fig. EV1A and "Flow cytometry and cell sorting" section for antibodies). cDC1s were CD11c$^+$ I-A/E$^+$ B220$^-$ XCR1$^+$ Sirpα$^-$ (see Fig. EV1A).

To generate effector OT-I cells (OT-I T$_E$ cells) (Buck et al, 2016), which are more sensitive than naïve OT-I as a readout for OVA cross-presentation by cDC1s, 100 U/mL recombinant mouse IL-2 (Peprotech) and 0.1 nM SIINFEKL (prepared by the Francis Crick Institute Chemical Biology STP) recognised by OT-I transgenic CD8$^+$ T cells in the context of H2-K$^b$ were added to RBC-lysed splenocytes from OT-I mice in R10$^+$ medium for 3 days. On days 3 and 4, cells were split 1:2 into complete R10$^+$ medium replacement containing 100 U/mL IL-2. OT-I T$_E$ cells were used on days 4 or 5. A similar method was used to prepare F5 T$_E$ cells using the ASNENDAM peptide (prepared by the Francis Crick Institute Chemical Biology STP), recognised by F5 transgenic CD8$^+$ T cells in the context of H2-D$^b$.

## Retroviral transduction

GP2-293 cells were seeded onto 10 cm dishes and transfected with 18 μL GeneJuice (Novagen), 6 μg pMD2.G, and 6 μg of pFB-IRES-

GFP plasmid coding for the protein of interest in 600 μL Opti-MEM medium (Thermo Fisher Scientific). Pseudotyped virus collected on days 1, 2, and 3 after transfection was 0.45 μm filtered and combined with 8 μg/mL polybrene (Sigma-Aldrich) to spinfect Phoenix-ECO cells grown in six-well plates for 90 min at 1000×$g$ at 30 °C. The medium was exchanged for fresh R10$^+$ medium. Then Phoenix-ECO were expanded and ecotropic viral supernatants from confluent cultures was 0.45 μm filtered and combined with 8 μg/mL polybrene to spinfect MuTuDCs and RAW 264.7 cells grown in six-well plates for 90 min at 1000×$g$ at 30 °C.

## Flow cytometry and cell sorting

Single cell suspensions were stained with LIVE/DEAD Fixable Blue or Aqua Dead Cell dye (Thermo Fisher Scientific) according to manufacturer's instructions with α-mouse CD16/32 or Fc block (clone 2.4G2, BD Pharmingen) and subsequently stained with fluorescent label-conjugated antibodies against the following mouse proteins: CD11c (clone N418, Biolegend), CD40 (clone 3/23, Biolegend), CD45R/B220 (clone RA3-6B2, Biolegend), CD80 (clone 16-10A1, Biolegend), CD83 (clone Michel-19, BD Biosciences), CD86 (clone GL-1, Biolegend; eBioscience), CD172a/Sirpα (clone P84, Biolegend), CD197/CCR7 (clone 4B12, BD Biosciences), DNGR-1/CLEC9A (clone 1F6, generated at the Francis Crick Institute, fluorochrome-labelled by Biolegend), Dectin-1/CLEC7A (clone 2A11, Bio-Rad), H2-Kb (clone AF6-88.5, Biolegend), H2-D$^d$ (clone 34-2-12, BD Pharmingen), H2-K$^d$/D$^d$ (clone 34-1-2S, Biolegend), I-A/E (clone M5/114.15.2, Thermo Fisher Scientific), and XCR1 (clone ZET, Biolegend) in PBS or FACS Buffer (PBS with 3% FCS and 2 mM EDTA ± 0.02% NaN$_3$).

For intracellular cytokine staining, cells were cultured for the last 4 h of stimulation at 37 °C in R10$^+$ medium containing GolgiPlug (BD Biosciences) and subsequently stained with α-mouse IL-12 p40 antibody (clone C15.6, Biolegend) using the CytoFix/CytoPerm kit (BD Biosciences) according to the manufacturer's instructions. Analytes released into supernatants from stimulated MuTuDCs and RAW 264.7 cells were quantified by cytokine bead array using LEGENDplex Mouse Macrophage/Microglia Panel (Biolegend) according to the manufacturer's instructions.

Cells were resuspended in FACS buffer (PBS with 3% FCS and 2 mM EDTA) and acquired live or fixed (Nordic-MUbio) on a LSRFortessa, LSRFortessa X-20, or FACSymphony (BD Biosciences). Cells were sorted in PBS with 2% FCS and 1 mM EDTA on a FACS Aria III, Fusion, Influx (BD Biosciences), or Avalon (Propel Labs) using a 100 μm nozzle into R10$^+$ medium. Data were analysed using FlowJo software version 10.

## DNGR-1L and bead preparations

DNGR-1L, comprising of F-actin-myosin II complexes, was prepared as previously described (Schulz et al, 2018). Lyophilised G-actin and myosin II (Cytoskeleton) were reconstituted at 10 mg/mL in sterile water and stored at -80 °C. 20 μM F-actin was generated by diluting G-actin to 1 mg/mL with G- and F-buffer (Cytoskeleton) to initiate polymerisation for ≥1 h at room temperature (RT). F-actin was mixed in a 1:1 molar ratio with myosin II for an additional hour at RT. PBS was used as a diluent for further assays.

DNGR-1L beads were generated similarly to DNGR-1L with some modifications. Biotinylated G-actin (Cytoskeleton) reconstituted at 1 mg/mL with sterile water was mixed in a 1:1 molar ratio with nonbiotinylated G-actin and polymerised with G- and F-buffer for ≥1 h at RT. Biotinylated F-actin was mixed in a 1:1 molar ratio with myosin II for an additional hour at RT to generate biotinylated DNGR-1L. 2 μm streptavidin non-fluorescent, yellow-green (Polysciences), or 1 μm red fluorescent NeutrAvidin (Thermo Fisher Scientific) beads were washed with 1% BSA in PBS at ≥10,000×$g$. Biotinylated DNGR-1L was added to beads for ≥30 min on ice before washing with 1% BSA in PBS at ≥10,000×$g$. α-DNGR-1 IgG-coupled beads were made by incubating 2 μm streptavidin non-fluorescent beads (Polysciences) washed with 1% BSA in PBS with 0.15 mg/mL rat biotinylated α-DNGR-1 monoclonal IgG (clone 7H11) or isotype (clone MAC49) for ≥30 min on ice. Beads were washed with 1% BSA in PBS at ≥10,000×$g$.

DNGR-1L-OVA beads were made by first incubating washed (with 1% BSA in PBS at ≥10,000×$g$) 2 μm streptavidin non-fluorescent beads with 2 mg/mL ovalbumin (OVA) biotinylated using the DSB-X-biotinylation kit (Thermo Fisher Scientific) for 30 min on ice. Beads were washed with 1% BSA in PBS at ≥10,000×$g$ before proceeding with additional coupling to DNGR-1L as described above. All bead suspensions were sonicated for 5 min before use to disperse beads.

## Stimulation with DNGR-1 or Dectin-1 agonists and other agents

BM-FLT3L cDCs, MuTuDCs, or RAW 264.7 cells were cultured overnight at 37 °C unless otherwise specified with the following stimuli alone or in combination with other pharmacological agents: DNGR-1L (see above), zymosan depleted of TLR agonists by hot alkali treatment (Zym-D; InvivoGen), poly(I:C) (InvivoGen), IFN-αA (R&D Systems), rat α-mouse DNGR-1 (clones 1F6 or 7H11) or isotype-matched irrelevant specificity control (clone MAC49) generated at the Francis Crick Institute, C3K (PKM2 inhibitor, Universal Biologicals), DASA-58 (PKM2 agonist, Merck Life Science), 3AC (SHIP1 inhibitor, MedChemExpress), RMC-4550 (SHP-2 inhibitor, MedChem Express). Unless otherwise stated, cultured supernatant analytes were quantified using mouse CCL17/TARC, CCL22/MDC, IL-12 p40, TNF-α DuoSet ELISA kits (R&D Systems) according to the manufacturer's instructions. For measuring the agonistic activity of DNGR-1L or cross-linking with α-DNGR-1 IgG, NFAT reporter B3Z cells were stimulated overnight. Cells were washed once with PBS and LacZ activity was measured by lysing cells in chlorophenol red-β-D galactopyranoside (CPRG) (Roche)-containing buffer. In all, 1–4 h later, optical density 595 ($OD_{595}$) was measured using optical density 655 ($OD_{655}$) as a reference.

## Dead cell preparation

Dead cells were generated by irradiating with 240 mJ/cm$^2$ UVC in PBS followed by overnight culture in RPMI 1640 lacking FCS. OVA-dead cells refer to either irradiated BRAF$^{V600E}$ melanoma cells incubated with 10 mg/mL OVA (Sigma-Aldrich) at 37 °C for 1 h and washed with PBS before use or irradiated HeLa cells stably expressing a F-actin-targeting affimer fused to OVA (O. Schulz, unpublished) (Lopata et al, 2018). Both gave similar results. NP-dead cells refer to HeLa cells stably expressing influenza NP

(NP$_{NT60}$ containing ASNENMDAM$_{366-373}$) protein mutated to prevent nuclear localisation and fused to LifeAct (O. Schulz, unpublished) (Riedl et al, 2008).

## Cross-presentation assay

0.5 or 1.0 × 10$^5$ BM-FLT3L cDC1s or MuTuDCs were seeded in 96-well U-bottomed plates, unless otherwise stated. OVA- or NP-dead cells, DNGR-1L-OVA coupled beads, OVA peptide (SIINFEKL) or NP peptide (ASNENMDAM) were added at the indicated ratios/concentrations for 4 h at 37 °C. OT-I T$_E$ or F5 T$_E$ cells were layered on top at a 2:1 or 1:1 ratio unless otherwise specified and co-cultured overnight at 37 °C with cDC1s. T cell derived IFN-γ in supernatants was measured by in-house ELISA. Briefly, 96-well high-affinity Nunc MaxiSorp plates (Thermo Fisher Scientific) were coated overnight with rat α-mouse IFN-γ IgG (clone R4-6A2, BD Biosciences, 8 μg/mL in 0.1 M sodium bicarbonate buffer) before extensively washing in 0.05% Tween-20 in PBS. Plates were blocked for 1 h with 3% FCS in PBS (blocking buffer), washed once more, and incubated with T cell culture supernatants and recombinant IFN-γ (Peprotech) standard curve samples for 2 h. After washing, plates were incubated with biotin rat α-mouse IFN-γ IgG (clone XMG1.2, BD Biosciences, 1 μg/mL in blocking buffer) for 2 h, then ExtrAvidin-Alkaline Phosphatase (Merk Life Science, 1:5000 in blocking buffer) for 30 min, before developing with SIGMAFAST p-Nitrophenyl substrate solution (Sigma-Aldrich), as per the manufacturer's instructions. Absorbances at 405 nm (IFN-γ signal) and 540 nm (background) were measured after 5-20 min on a Spark plate reader (Tecan) and absolute concentrations determined using the standard curve. All ELISA steps were performed at RT.

## Phagocytosis assays

Necrophagy: prior to irradiation, BRAF$^{V500E}$ melanoma cells were labelled with Cell Tracker-Deep Red (CT-DR) dye (Thermo Fisher Scientific) for 1 h at 37 °C. CT-DR-labeled dead cells were added to BM-FLT3L cDCs or MuTuDCs at a series of ratios for 3–4 h at 37 °C. As a negative control, MuTuDCs were pre-incubated with 1 μM cytochalasin D (Sigma-Aldrich), which inhibits actin-polymerisation required for uptake, before addition of the highest ratio of dead corpses. Following target incubation, phagocytes were analysed by flow cytometry. LIVE/DEAD Fixable Blue or Aqua Dead Cell Dye (Thermo Fisher Scientific) was used to exclude non-internalised dead cell material.

Bead uptake: phagocytosis was also assessed by flow cytometry in MuTuDCs with DNGR-1L coupled beads added at different ratios for 3–4 h. Fluorochrome-conjugated streptavidin staining was used to exclude non-phagocytosed material by binding to biotinylated F-actin exposed on non-internalised beads.

## CRISPR/Cas9 gene editing

Genetic deletion in MuTuDCs was performed as previously described (Freund et al, 2020; Henry et al, 2023a). Briefly, 25 μg recombinant Cas9 nuclease V3 and 40 μM target sgRNA (IDT) were complexed in the presence of 1.6 μM IDT Alt-R Cas9 Electroporation Enhancer per reaction for 25 min at RT. Two sgRNA guides were used per gene. In total, 2 × 10$^6$ MuTuDCs were mixed with each RNP complex in primary nucleofection solution

(P3) in a 1.5 mL Eppendorf and transferred to a Nucleocuvette. The RNP loaded cells were electroporated with the CM-137 programme in a 4D Nucleofector (Lonza) before transferring to prewarmed R10$^+$ medium without antibiotics for initial culture. Efficiency of CRISPR-deletion was routinely nearly complete as assessed at the protein level by Western blot from bulk populations. Experiments were repeated with single cell clones with KO confirmed by western blot.

## Confocal microscopy

4–5 × 10$^5$ MuTuDCs were seeded onto glass coverslips (18 mm, VWR) the day prior to the experiment. The cells were then incubated with α-DNGR-1 IgG-coupled beads at 5:1 ratio and then again 1.5 h later for a total of 3 h at 37 °C. All subsequent steps were performed at RT. Cells were fixed with 4% paraformaldehyde, washed with PBS, and quenched with 0.5 mL 50 mM NH$_4$Cl for 5–10 min and washed again with PBS. Samples were blocked for 20 min with 2% BSA in PBS (blocking buffer). If staining for non-internalised beads, α-rat IgG AF647 was added 1:200 in PBS for 30 min (Thermo Fisher Scientific). After washing with PBS, cells were permeabilised with 0.1% Triton-X 100 in PBS for 15 min. Cells were washed with blocking buffer and then blocked for 20 min. Then 0.165 μM Alexa Fluor Plus 405 Phalloidin (Thermo Fisher Scientific), rabbit α-mouse monoclonal SYK pY519/20 (clone C87C1, CST) or polyclonal SHIP1 pY1021 (STEMCELL Technologies) antibodies (1:200) were incubated on coverslips for 1 h. Next, samples were washed with blocking buffer and stained with α-rat IgG AF647 (Thermo Fisher Scientific) at 1:200 for 1 h. Lastly, coverslips were washed with blocking buffer then PBS and then mounted onto glass slides (Sail Brand) using Prolong Diamond Antifade Mountant (Thermo Fisher Scientific). Samples were imaged on a Zeiss LSM880 inverted microscope. Image processing and analysis was performed using FIJI and Zeiss ZEN Black and Blue software.

## Extracellular flux analysis

Oxygen consumption rates (OCR) and extracellular acidification rates (ECAR) were measured in XF media (non-buffered RPMI 1640 containing 25 mM glucose, 2 mM glutamine, 1 mM sodium pyruvate, pH = 7.4) under basal conditions and in response to DNGR-1L, Zym-D, C3K, DASA-58, 1 μM oligomycin, 1.5 μM fluoro-carbonyl cyanide phenylhydrazone (FCCP) and 100 nM rotenone + 1 μM antimycin A (Sigma-Aldrich) using a 96-well Agilent Seahorse XF Analyzer as previously described (van der Windt et al, 2016).

## Metabolite extraction and LC-MS

MuTuDCs were seeded at 5 × 10$^6$ per 10 cm plate the day before in R10$^+$ medium. The culture supernatant was replaced with glucose-free R10$^+$ medium containing 10 mM $^{13}$C uniformly labelled glucose (Cambridge Isotope Laboratories) ± 12.5 nM DNGR-1L. Cells were collected immediately (time 0) and subsequently at 0.5, 1, and 2 h post-stimulation. Harvested cells were washed with ice-cold PBS before being metabolically quenched by rapidly cooling on a dry ice/ethanol slurry. Metabolites were extracted by resuspending the cell pellet in ice-cold HPLC-grade methanol/

chloroform (2:1, v/v) and incubated at 4 °C with three sonication steps (8 min each) within an hour. Samples were centrifuged for 10 min at 16,000×$g$ (4 °C) and supernatants collected. The remaining sample was re-extracted using ice-cold HPLC-grade methanol/water (2:1, v/v), sonicated for 8 min (4 °C), and centrifuged for 10 min at 16,000×$g$ (4 °C). The combined extraction solvents were dried in a glass insert placed in a LC-MS vial (Agilent, 5182-0716) using a SpeedVac (Christ RVC 2-33 CDplus). In all, 300 μL of metabolite extraction buffer containing 5 μM $^{13}$C,$^{15}$N-valine (used as an internal standard) was added to each dried sample, and stored at −80 °C.

Metabolite analysis was performed by LC-MS using a Q-Exactive Plus (Orbitrap) mass spectrometer (Thermo Fisher Scientific) coupled with a Vanquish UHPLC system (Thermo Fisher Scientific). The chromatographic separation was performed on a SeQuant Zic pHILIC (Merck Millipore) column (5 μm particle size, polymeric, 150 × 4.6 mm) using a gradient program at a constant flow rate of 300 μL/min over a total run time of 25 min. The elution gradient was programmed as a decreasing percentage of solvent B from 80% to 5% over 17 min, holding at 5% B for 3 min, and finally re-equilibrating the column at 80% B for 4 min. Solvent A was 20 mM ammonium carbonate solution in water supplemented by 4 mL/L of a solution of ammonium hydroxide (35% in water), and solvent B was acetonitrile. MS was performed with positive/negative polarity switching using a Q-Exactive Orbitrap (Thermo Fisher Scientific) with a HESI II probe. Parameters were as follows: spray voltage 3.5 and 3.2 kV for positive and negative modes, respectively; probe temperature 320 °C; observed sheath and auxiliary gases at 30 and 5 arbitrary units, respectively; and observed full scan range at 70–1050 $m/z$, with settings of AGC target and resolution as balanced ($3 \times 10^6$) and high (70,000), respectively. Data were recorded using Xcalibur 4.2.47 software (Thermo Fisher Scientific). Mass calibration was performed for both ESI polarities before analysis using the standard Thermo Fisher Scientific Calmix solution. To enhance calibration stability, lock-mass correction was also applied to each analytical run using ubiquitous low-mass contaminants. Parallel reaction monitoring (PRM) acquisition parameters were as follows: resolution was 17,500, and collision energies were set individually in high-energy collisional dissociation (HCD) mode. Metabolites were identified and quantified by accurate mass and retention time and by comparison with the retention times, mass spectra, and responses of known amounts of authentic standards using TraceFinder 4.1 EFS software (Thermo Fisher Scientific). Label incorporation and abundance was measured using TraceFinder 4.1 EFS software. Label incorporation into individual metabolites was estimated as the percentage of the metabolite pool containing one or more $^{13}$C atoms after correction for natural abundance isotopes. Abundance was calculated relative to the internal standard.

Samples acquired by GC-MS were speed vacuum dried after addition of 1 nmol *scyllo*-inositol (used as an internal standard). Extracts were washed twice with methanol. Data acquisition was performed largely as previously described (Mendez-Lucas et al, 2020), using an Agilent 7890B-5977A GC-MSD in EI mode after derivatisation of dried extracts by addition of 20 μL methoxyamine hydrochloride (20 mg/mL in pyridine (both Sigma), RT, >16 h) and 20 μL BSTFA + 1% TMCS (Sigma, RT, >1 h). GC-MS parameters were as follows: carrier gas, helium; flow rate, 0.9 mL/min; column, DB-5MS (Agilent); inlet, 270 °C; temperature gradient, 70 °C

(2 min), ramp to 295 °C (12.5 °C/min), ramp to 320 °C (25 °C/min, 3 min hold). Scan range was m/z 50-550. Data analysis was performed using MANIC software version 3.0.18, an in house-developed adaptation of the GAVIN package (Behrends et al, 2011). Pyruvate and lactate were identified and quantified by comparison to authentic standards, and label incorporation was estimated as the percentage of the metabolite pool containing ≥1 $^{13}$C atoms after correction for natural abundance.

## Western blotting

Cell lysates were prepared using Laemmli sample buffer (Bio-Rad) supplemented with 2-mercaptoethanol and HALT Protease and Phosphatase Inhibitor Cocktail (Thermo Fisher Scientific), boiled for 7 min at 95 °C, resolved by SDS-PAGE with 4-20% Mini-PROTEAN TGX Precast Protein gels (Bio-Rad), and transferred onto 0.2 μm PVDF membranes using the Trans-Blot Turbo Transfer System (Bio-Rad) according to the manufacturer's instructions. Membranes were blocked for 1 h with 5% w/v milk and 0.1% Tween-20 in TBS (TBST) and incubated with the following α-mouse monoclonal antibodies (1:1000) in 5% w/v milk or BSA in TBST overnight at 4 °C: IκBα (clone 44D4), p38 pT180/pY182 (clone D3F9), p44 pT202/pY204 (clone 9101), SHP-1 pY564 (clone D11G5), SHP-2 pY542 (clone 3751), SHP-2 pY580 (clone 3703), SYK pY352 (clone 65E4), and β-actin-HRP (clone 13E5) from Cell Science Technologies (CST), and SHIP1 pY1021 (polyclonal) from STEMCELL Technologies. Primary antibody incubations were followed by incubation with secondary HRP-conjugated antibody (1:2000–5000, CST) in 5% w/v milk in TBST for 1 h and visualised using an ImageQuant 800 (Amersham). Densities were acquired using FIJI (NIH) and calculated relative to β-actin loading control signals.

## RT-qPCR and RNAseq analysis

RNA was extracted using the QIAshredder and RNeasy kit (Qiagen) according to manufacturer's instructions. Single-strand cDNA was synthesised by first incubating the isolated RNA with 0.25 μg random primers (Thermo Fisher Scientific) at 72 °C for 10 min, followed by incubation with 1× Superscript II buffer, 10 nM DTT, 0.5 mM dNTPs and 0.5 mL reverse transcriptase Superscript II (Thermo Fisher Scientific) at 42 °C for 1 h, 70 °C for 15 min and then kept at 4 °C. All real-time (RT)-qPCR was performed with Taqman qPCR primers and universal PCR master mix using QuantStudio 3, 5, or 7 RT-PCR systems (Thermo Fisher Scientific). Expression levels of mRNA were normalised to the expression of a housekeeping gene (β-actin).

For bulk RNAseq, RNA was extracted as above. Biological replicate libraries were prepared using the NEBNext Ultra II Directional PolyA mRNA kit and sequenced on Illumina HiSeq 4000 platform, generating ~25 million 100 bp paired end reads per sample. The RSEM package (version 1.3.30) in conjunction with the STAR alignment algorithm (version 2.5.2a) was used for the mapping and subsequent gene-level counting of the sequenced reads with respect to Ensembl mouse GRCm.38.89 version transcriptome. Normalization of raw count data and differential expression analysis was performed with the DESeq2 package (version 1.24.1) within the R programming environment (version 4.1.0). Differential analysis was done using DESEQ2 Wald's test.

False Discovery Rate (FDR) corrected p-value < 0.05 was used to threshold for significance in the differential gene expression analysis. Gene Set Enrichment Analysis (GSEA) to rank differentially expressed genes against the molecular signatures database for gene-ontology terms was used to identify pathways enriched in cells from different genotypes and treatment conditions.

## Pull-down and mass spectrometry

DNGR-1 KO MuTuDCs were treated with 100 μM Na$_3$VO$_4$ with 0.0006% v/v H$_2$O$_2$ for 15 min at 37 °C in R10$^+$ medium. Cells were washed and harvested using ice-cold PBS and then lysed with a pull-down lysis buffer composed of Cell Lysis Buffer (CST) supplemented with 1 mM PMSF and Halt Protease and Phosphatase Inhibitor Cocktail (Thermo Fisher Scientific). Lysates were pre-cleared with Pierce streptavidin magnetic beads (Thermo Fisher Scientific) that had been washed with pull-down wash buffer (50 mM Tris, 10 mM NaCl, 0.5 mM EDTA, 1 mM Na$_3$VO$_4$, 1% Brij-58, and Halt Protease and Phosphatase Inhibitor Cocktail). In all, 20 μg of biotinylated peptides composed of the unphosphorylated and phosphorylated intracellular domains of WT and I6G DNGR-1 conjugated to pegylated biotin were incubated on rotation with the pre-cleared lysates for 1.5 h at 4 °C. For pull-down, washed streptavidin magnetic beads were then added for an additional 1.5 h on rotation at 4 °C. Peptide-associated beads were washed using pull-down wash buffer placed in a magnetic holder and stored in 50 mM ammonium bicarbonate at 4 °C until further processing.

Proteins bound to DNGR-1 peptides were reduced with 10 mM DTT and alkylated with 55 mM iodoacetamide. After alkylation, proteins were digested with 6 ng/mL trypsin (Promega, UK) overnight at 37 °C. The resulting peptides were extracted in 2% v/v formic acid, 2% v/v acetonitrile, and evaporated dry in a speed vac. Prior to analysis the samples were solubilised in 20 μL of 0.1% v/v trifluoroacetic acid. Digests were analysed by nano-scale capillary LC-MS/MS using an Ultimate U3000 HPLC (Thermo Fisher Scientific) to deliver a flow of approximately 250 nL/minute. A C18 Acclaim PepMap100 3 mm, 75 mm×20 mm nanoViper (Thermo Fisher Scientific), trapped the peptides prior to separation on an EASY-Spray PepMap RSLC 2 μm, 100 Å, 75 μm × 500 mm nanoViper column (Thermo Fisher Scientific). Peptides were eluted with a 60 min gradient of acetonitrile (2–80%). The analytical column outlet was directly interfaced via a nano-flow electrospray ionisation source, with an Orbitrap Eclipse Tribrid (Thermo Fisher Scientific) mass spectrometer which was operated in positive ionisation mode to acquire data. Data were collected in a data-dependent acquisition mode, with instrument settings: MS1 data acquired in the Orbitrap at a resolution of 120k, standard AGC target, 50 ms maximum injection time, dynamic exclusion of ± 10 ppm and 60 s, a mass range of 330–1500 m/z and profile mode data capture. MS2 data were acquired in the ion trap using a 1.2 m/z isolation window, standard AGC target, maximum injection time set to Dynamic, HCD of 30% collision energy, 1 ms activation time and centroid mode data capture.

All raw files were processed with MaxQuant v2.0.1.0 using default settings and searched against the UniProt KB *Mus Musculus* with the Andromeda search engine integrated into the MaxQuant software suite. Enzyme search specificity was Trypsin/P. Up to two missed cleavages for each peptide were allowed.

Carbamidomethylation of cysteines was set as fixed modification with oxidized methionine and protein N-acetylation considered as variable modifications. The false discovery rate was fixed at 1% at the peptide and protein level. Statistical analysis was carried out using the Perseus module (v1.6.14.0) of MaxQuant. Prior to statistical analysis, peptides mapped to known contaminants, reverse hits, and protein groups only identified by site were removed.

## Quantification and statistical analysis

All statistical analyses were performed using GraphPad Prism software version 10. Comparisons for two groups were calculated using unpaired two-tailed Student's *t* test. Comparisons for two or more groups were done by one- or two-way ANOVA followed by Tukey multiple comparisons post-hoc correction. Data are plotted as mean ± SEM or ± SD as stated in figure legends. All error bars are shown: where they appear to be missing, they are too small to be visible. The following scheme was used to represent statistical significance: $*P < 0.05$, $**P < 0.01$, $***P < 0.001$, $****P < 0.0001$. For BM-FLT3L cDC1 cultures, biological replicates refer to independent cultures from different mice. For MuTuDC and RAW264.7 cells, biological replicates refer to independently treated cell batches.

## Data availability

The metabolomics datasets produced in this study are available at the following database: MetaboLights repository with the accession number MTBLS1457. The RNAseq datasets produced in this study are available at the following database: NCBI's GEO repository under the identifier GSE287030. The proteomics datasets produced in this study are available at the following database: ProteomeXchange Consortium via the PRIDE partner repository with the dataset identifier PXD059637.

The source data of this paper are collected in the following database record: biostudies:S-SCDT-10_1038-S44318-025-00620-z.

## Peer review information

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

## Acknowledgements

We thank past and present members of the Immunobiology Laboratory for helpful discussion and suggestions including Adi Biram, Bruno Federico, Cécile Piot, and Francesca Gasparrini for technical advice and help. We thank the Cell Services, Chemical Biology (Nicola O'Reilly, Dhira Joshi, Stefania Federico), Flow Cytometry (Hefin Rhys, Sukhveer Purewal, Ana Água-Doce, Kerol Bartolovic, Debripriya Das, Sina Namjou, Steven Lim), and Genomics (Jerome Nicod, Ashley Fowler, Deb Jackson, Marg Crawford, Maria Rodriguez) Science

Technology Platforms of the Francis Crick Institute for their support throughout this project. This work was supported by the Francis Crick Institute, which receives core funding from Cancer Research UK (CC2090), the UK Medical Research Council (CC2090), and the Wellcome Trust (CC2090); an European Research Council Advanced Investigator grant (AdG 268670), Wellcome Investigator Awards (106973/Z/15/Z and 223136/Z/21/Z), and a prize from the Louis-Jeantet Foundation to CRS; a Boehringer Ingelheim Fonds Fellowship to WS; and H2020 Marie Sklodowska-Curie Actions Individual Fellowships awarded to MDB and CMH (837951 and 792770, respectively). The synopsis image was created using Biorender.com.

## Author contributions

**Michael D Buck**: Conceptualization; Resources; Data curation; Formal analysis; Funding acquisition; Validation; Investigation; Visualization; Methodology; Writing—original draft; Project administration; Writing—review and editing. **Tomás Castro-Dopico**: Formal analysis; Validation; Investigation; Methodology; Writing—review and editing. **Oliver Schulz**: Formal analysis; Investigation; Methodology; Writing—review and editing. **Ana Cardoso**: Formal analysis; Validation; Investigation; Methodology; Writing—review and editing. **Probir Chakravarty**: Formal analysis; Validation; Investigation; Methodology; Writing—review and editing. **Nathalie Legrave**: Formal analysis; Validation; Investigation; Methodology; Writing—review and editing. **Conor M Henry**: Formal analysis; Validation; Investigation; Methodology; Writing—review and editing. **Johnathan Canton**: Formal analysis; Validation; Investigation; Methodology; Writing—review and editing. **Estelle Wu**: Formal analysis; Validation; Investigation; Methodology; Writing—review and editing. **Sonia Lee**: Resources; Methodology; Project administration; Writing—review and editing. **Neil C Rogers**: Resources; Methodology; Project administration; Writing—review and editing. **Enzo Z Poirier**: Formal analysis; Validation; Investigation; Methodology; Writing—review and editing. **William Stainier**: Formal analysis; Validation; Investigation; Methodology; Writing—review and editing. **Victor Bosteels**: Formal analysis; Validation; Investigation; Methodology; Writing—review and editing. **Eleanor Childs**: Formal analysis; Validation; Investigation; Methodology; Writing—review and editing. **James I MacRae**: Formal analysis; Validation; Investigation; Methodology; Writing—review and editing. **J Mark Skehel**: Formal analysis; Validation; Investigation; Methodology; Writing—review and editing. **Santiago Zelenay**: Resources; Writing—review and editing. **Caetano Reis e Sousa**: Conceptualization; Supervision; Funding acquisition; Writing—original draft; Project administration; Writing—review and editing.

Source data underlying figure panels in this paper may have individual authorship assigned. Where available, figure panel/source data authorship is listed in the following database record: biostudies:S-SCDT-10_1038-S44318-025-00620-z.

## Funding

## Disclosure and competing interests statement

CRS is a founder of Adendra Therapeutics and owns stock options and/or is a paid consultant for Adendra Therapeutics, Montis Biosciences, and Bicycle Therapeutics, all unrelated to this work. CRS also holds appointments as Visiting Professor at Imperial College London and at King's College London and as honorary professor at University College London. CRS is also a member of the Advisory Editorial Board of *The EMBO Journal*. This has no bearing on the editorial consideration of this article for publication. The authors declare no competing interests.

# Expanded View Figures

**Figure EV1.   Assessment of DNGR-1 stimulation on cDC1 activation (related to Fig. 1).**

(**A**) Representative flow cytometry gating strategy for analyzing cDCs derived from bone marrow-FLT3L cultures (BM-FLT3L) before and after XCR1$^+$ MACS enrichment. (**B**) DNGR-1 expression by WT BM-FLT3L cDC1 and cDC2 compared to C9$^{KI-Cre}$ cDC1. (**C–E**) WT, C9$^{KI-Cre}$ BM-FLT3L cDC1s or (**F, G**) C9 KO MuTuDCs or those reconstituted with indicated receptors were cultured overnight ± designated stimuli and assessed for surface expression of the specified markers by flow cytometry. (**C**) H2-K$^b$ expression from biological duplicates combined from two experiments with mean ± SEM plotted (left) and representative histograms (right). (**D**) Representative histograms of staining for indicated markers. (**E**) Cells were cultured overnight on uncoated plates or plates coated with different concentrations of α-DNGR-1 IgG (clone 1F6 or 7H11, as indicated) and assessed for indicated surface marker expression by flow cytometry. MFI values from biological duplicates pooled from two independent experiments with mean ± SEM is plotted. (**F**) Representative histograms of staining for indicated markers. (**G**) H2-K$^b$ expression from biological duplicates pooled from two independent experiments with mean ± SEM is plotted. Data are representative of two (**A–G**) independent experiments. (**C, E, G**) Data were analysed using Tukey-corrected two-way ANOVA with significant values comparing against untreated samples plotted. (**C**) ****$P < 0.0001$.

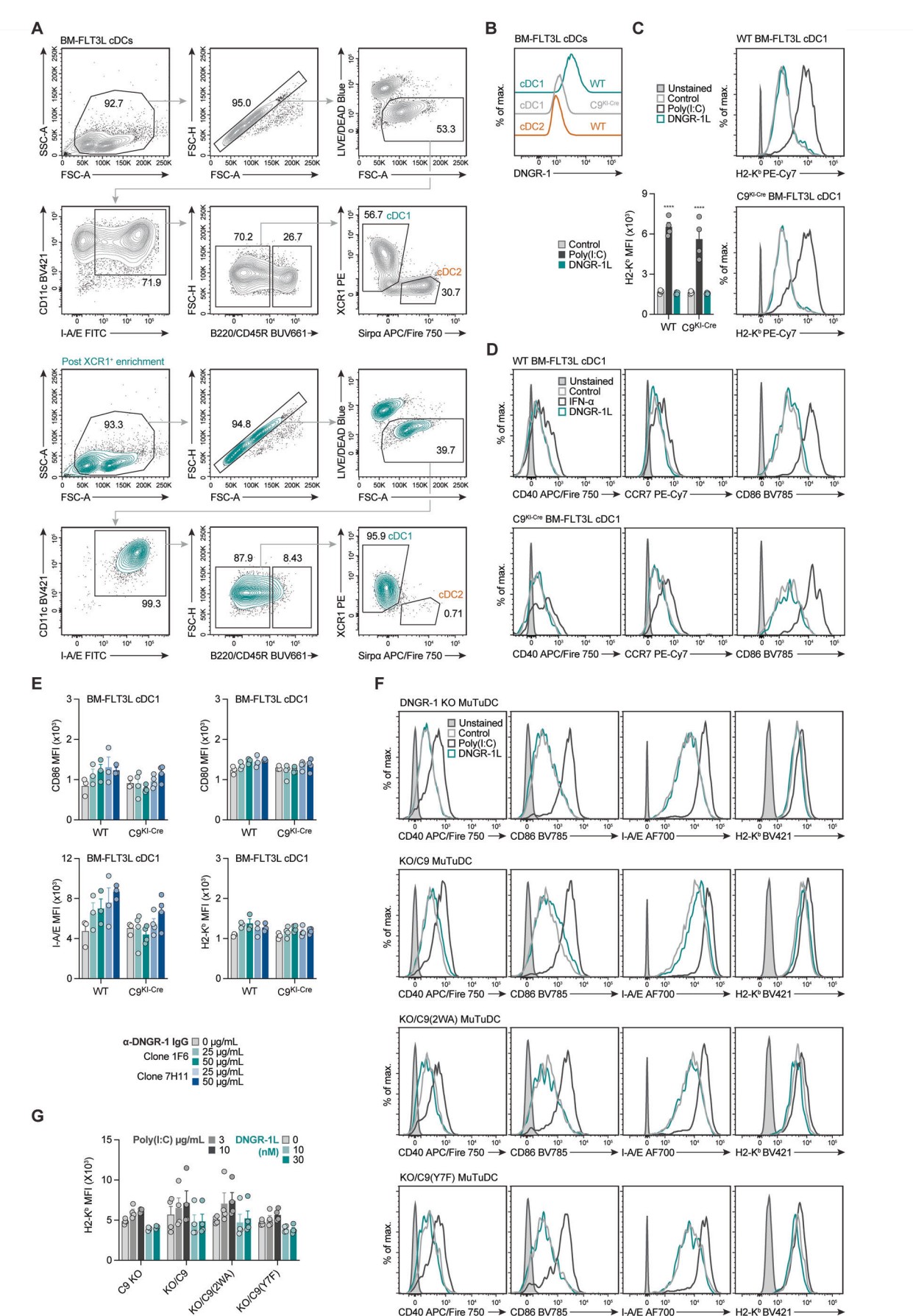

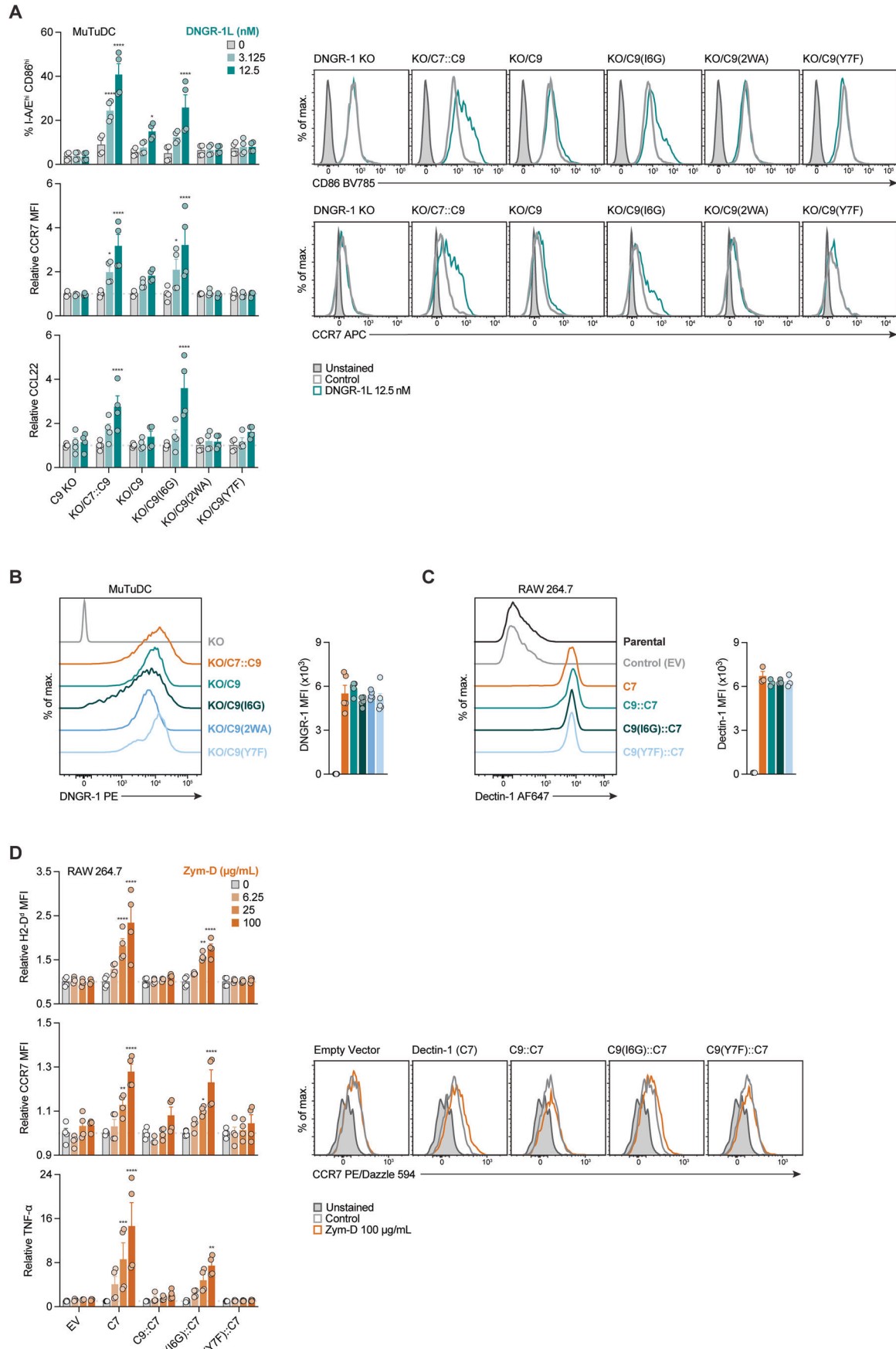

**Figure EV2.  Cell lines to analyse DNGR-1 function (related to Fig. 2).**

(A) Analysis of the indicated surface marker expression (top, middle; flow cytometry) or CCL22 released into cultured supernatants (bottom; ELISA) from C9 KO MuTuDCs reconstituted or not with the indicated receptors and stimulated overnight ± DNGR-1L. Mean ± SEM from biological replicates pooled from two independent experiments (left) and representative flow cytometry profiles (right) are plotted. Data here are partly represented in Fig. 2D,E and are relative to average of untreated controls to better emphasise the response to DNGR-1L. Dotted line represents 1. (B) Flow cytometric analysis of surface DNGR-1 (C9) expression in C9 KO MuTuDCs or those reconstituted or not with the indicated receptors. Cell lines were established after sorting for equal expression of DNGR-1. (C) Flow cytometric analysis of surface Dectin-1 (C7) expression by parental RAW 264.7 cells or cells ectopically expressing the indicated receptors or transduced with empty vector (EV). Cell lines were established after sorting for equal expression of Dectin-1. (B, C) Representative histograms (left) and mean MFI ± SEM (right) are plotted from biological (B) quintuplets or (C) triplicates. (D) Analysis of the indicated surface marker expression (top, middle; flow cytometry) or TNF-α released into cultured supernatants (bottom; ELISA) from RAW 264.7 cells ectopically expressing the indicated receptors or transduced with EV and stimulated overnight ± Zym-D. Mean ± SEM from biological replicates pooled from two independent experiments (left) and representative flow cytometry profiles (right) are plotted. Data here are partly represented in Fig. 2F and are relative to average of untreated controls to better emphasise the response to Zym-D. Dotted line represents 1. Data are representative of two (D), or three (A–C) independent experiments. Data were analysed using Tukey-corrected two-way ANOVA with significant values comparing against untreated samples plotted (A, D). (A) I-A/E$^{hi}$ CD86$^{hi}$ *$P = 0.0117$, ****$P < 0.0001$; CCR7 *$P = 0.0279$ (KO/C7::C9), $P = 0.0134$ (KO/C9(I6G)), ****$P < 0.0001$; CCL22 ****$P < 0.0001$, (D) H2-D$^d$ **$P = 0.0011$, ****$P < 0.0001$; CCR7 *$P = 0.0319$, **$P = 0.0040$, ****$P < 0.0001$; TNF-α **$P = 0.0033$, ***$P = 0.0004$, ****$P < 0.0001$.

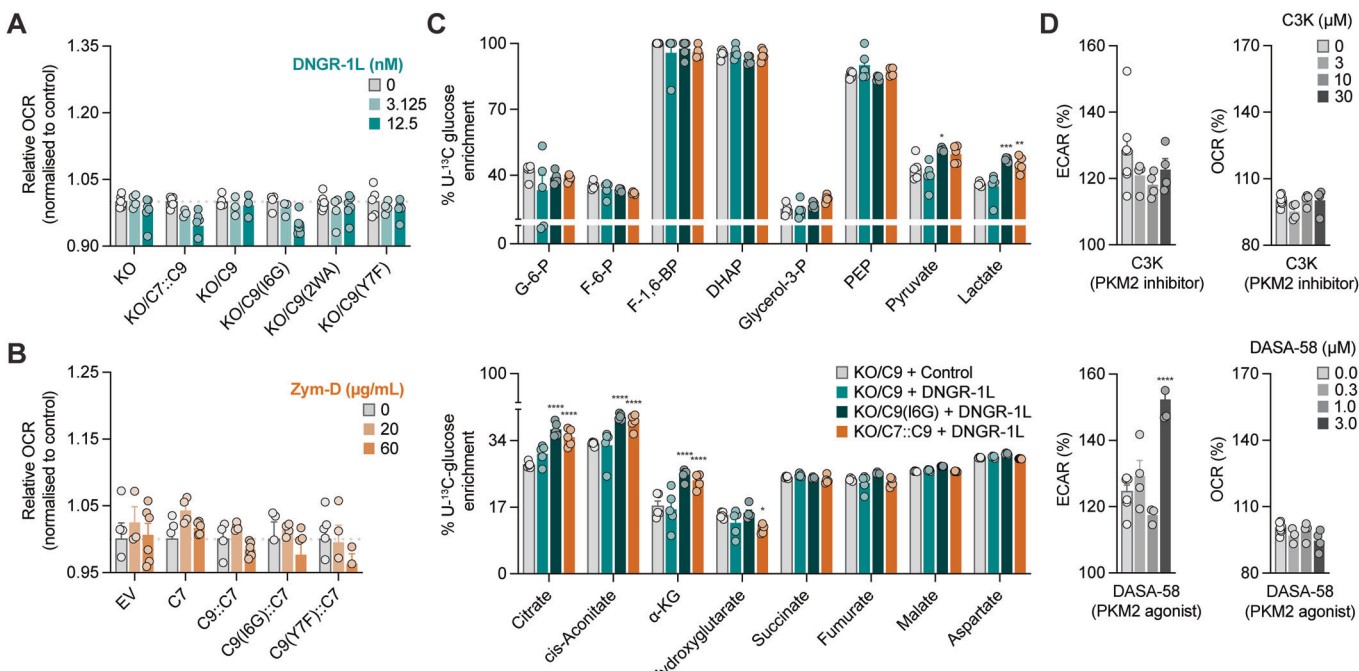

**Figure EV3.  Assessment of metabolism changes induced by DNGR-1 signalling (related to Fig. 3).**

(A, B) Oxygen consumption rates (OCR) measured at 2 h after ± DNGR-1L injection of (A) C9 KO MuTuDCs or those reconstituted with the indicated receptors or (B) RAW 264.7 cells ectopically expressing the indicated receptors or EV injected ± Zym-D, $n = 3$–6 per group. Data normalised to baseline measurement immediately after injection with stimuli. Mean ± SEM relative to untreated samples is plotted. (C) Fractional labelling of glycolytic (top) or tricarboxylic acid cycle (bottom) metabolites in C9 KO MuTuDCs reconstituted with indicated receptors and stimulated ± 12.5 nM DNGR-1L cultured with uniformly-labelled U-$^{13}$C-glucose introduced at the time of stimulation. Data shown as mean ± SEM from five biological replicates. (D) Extracellular acidification rate (ECAR) and OCR of MuTuDCs treated for 2 h with C3K (PKM2 inhibitor) or DASA-58 (PKM2 agonist). Data normalised as in (A). Mean ± SEM is plotted ($n = 4$ per treatment and 9 for untreated samples). Data are representative of two (A, B, D) independent experiments. Data were analysed using Tukey-corrected two-way ANOVA with significant values comparing against untreated controls plotted (A–D). (C) (top) *$P = 0.0128$, **$P = 0.0084$, ***$P = 0.0006$; (bottom) *$P = 0.0136$, ****$P < 0.0001$, (D) ****$P < 0.0001$.

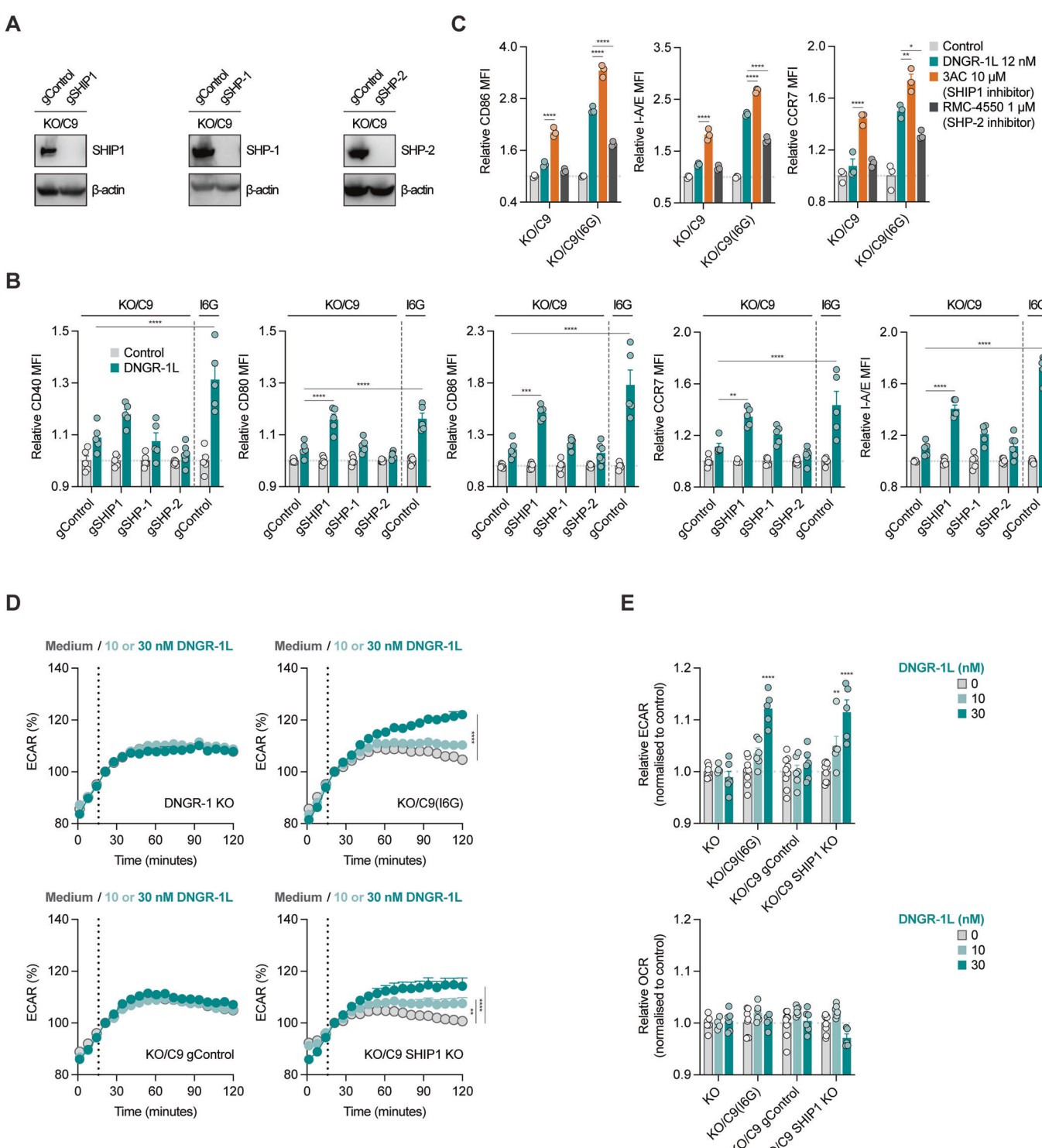

◀  **Figure EV4.  SHIP1 inhibition rescues DNGR-1 mediated cDC1 activation. (related to Fig. 4).**

(A) Western blot analysis of C9 KO MuTuDCs reconstituted with C9 made deficient for target proteins using CRISPR/Cas9. Two guide (g) RNAs that target SHIP1 (gSHIP1), SHP-1 (gSHP-1), SHP-2 (gSHP-2) or scrambled sequences (gControl) were used to complex with recombinant Cas9 protein for nucleofection. (B) Flow cytometric analysis of surface marker MFIs detected from KO/C9, KO/C9(I6G) or KO/C9 SHIP1, SHP-1, or SHP-2 deficient (KO) MuTuDCs cultured overnight ± 12.5 nM DNGR-1L. Mean ± SEM relative to untreated (Control) samples from biological replicates pooled from two independent experiments is plotted. (C) Flow cytometric analysis of surface protein MFIs from C9 KO MuTuDCs reconstituted with C9 or (I6G) MuTuDCs stimulated ± 12.5 nM DNGR-1L alone or in the presence of SHIP1 inhibitor 3AC or SHP-2 inhibitor RMC-4550 overnight. Mean ± SEM relative to untreated (Control) samples from biological triplicates is plotted. (D, E) Extracellular acidification rate (ECAR, indicator of glycolysis) measured at baseline and after ± 10 or 30 nM DNGR-1L injection of C9 KO SHIP1 sufficient and deficient MuTuDCs reconstituted with indicated receptors. $N = 4$–9 per group. (D) Data normalised to baseline measurement immediately after injection with stimuli and shown as % of baseline. Mean ± SEM is plotted. (E) ECAR at 2 h post-treatment. Data normalised as in (C). Mean ± SEM is plotted. Data are representative of one (C) or two (A, B, D, E) independent experiments. Data were analysed using Tukey-corrected two-way ANOVA. Comparisons are indicated (B, C) or against untreated controls (D, E) with significant values plotted. (B) **$P = 0.0011$, ****$P < 0.0001$, (C) *$P = 0.0177$, **$P = 0.0029$, ****$P < 0.0001$, (D) **$P = 0.0071$, ****$P < 0.0001$, (E) **$P = 0.0071$, ****$P < 0.0001$.

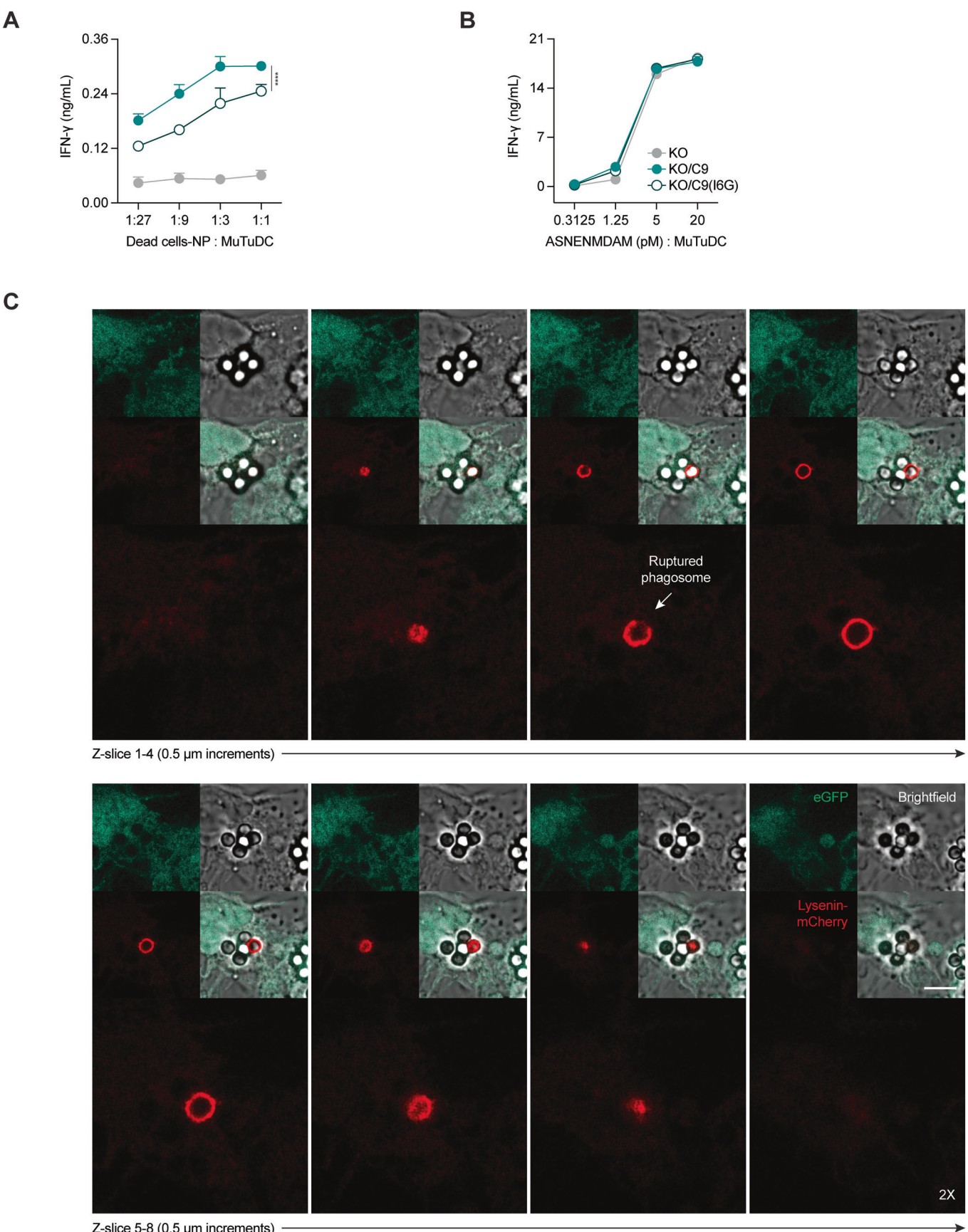

**A**

IFN-γ (ng/mL)

Dead cells-NP : MuTuDC

**B**

IFN-γ (ng/mL)

ASNENMDAM (pM) : MuTuDC

KO
KO/C9
KO/C9(I6G)

**C**

Ruptured phagosome

Z-slice 1-4 (0.5 μm increments)

eGFP | Brightfield

Lysenin-mCherry

2X

Z-slice 5-8 (0.5 μm increments)

◄ **Figure EV5. Cross-presentation of NP antigen and phagosomal rupture (related to Fig. 5).**

(A, B) IFN-γ release from F5 $T_E$ cells co-cultured with C9 KO MuTuDCs reconstituted or not with the indicated receptors and incubated with (A) NP-expressing dead cells, or (B) ASNENMDAM peptide. Mean ± SD from biological triplicates is plotted. (C) Confocal microscopic analysis of lysenin-mCherry fusion protein-expressing KO/C9 MuTuDCs co-cultured with α-DNGR-1 IgG-coupled beads. 0.5 μm increments of consecutive Z-slices of the same image in Fig. 5H. Arrow indicates ruptured area of lysenin-mCherry$^+$ phagosome indicated by loss of mCherry signal. Scale bar = 2 μm. Data are representative of two independent experiments (A–C). Data were analysed using Tukey-corrected two-way ANOVA. Comparisons are indicated (A, B) with significant values plotted. (A) ****$P < 0.0001$.

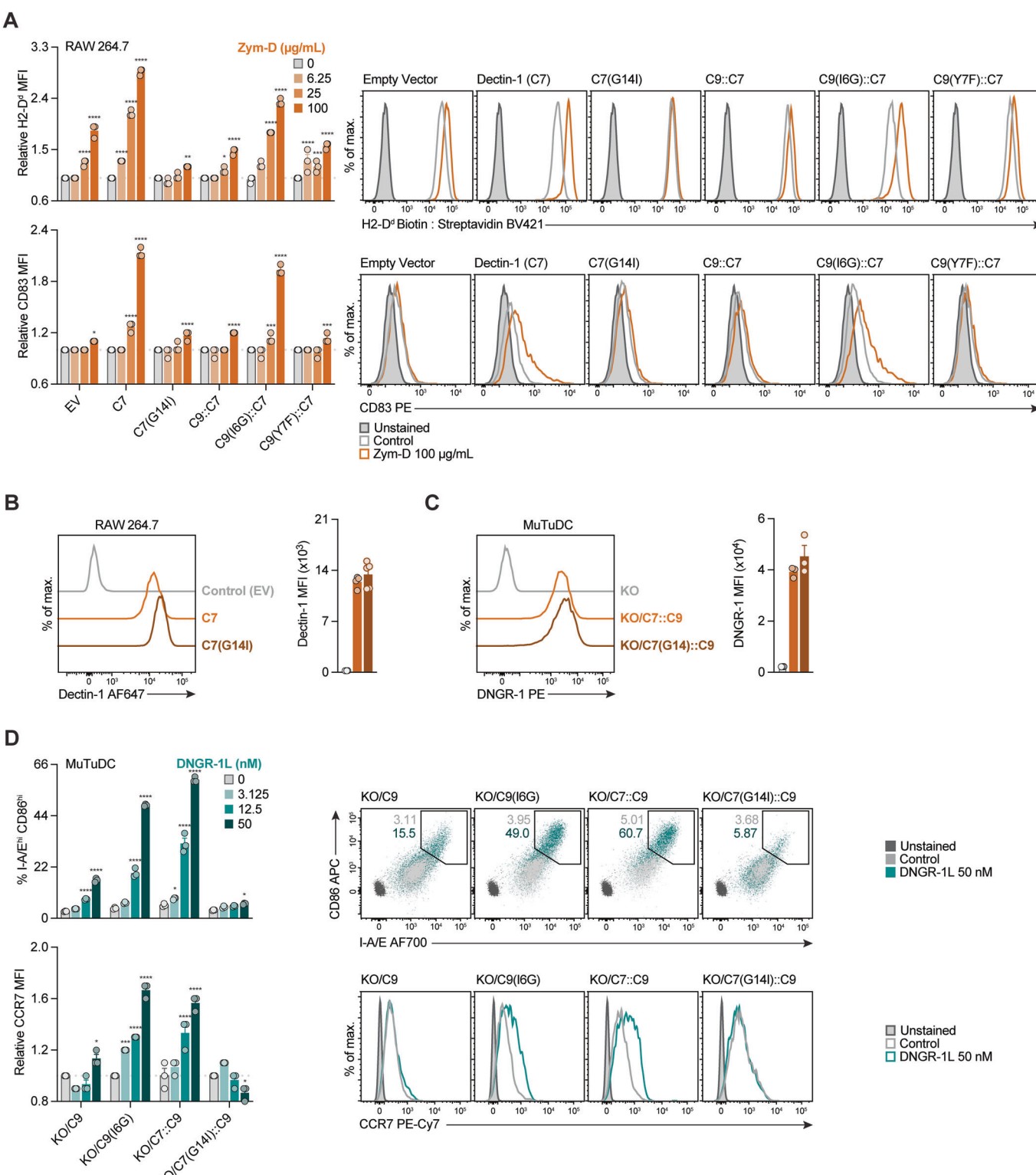

**Figure EV6.  Cell lines to analyse Dectin-1 function (related to Fig. 6).**

(A) Flow cytometric analysis of the indicated surface marker expression by RAW 264.7 cells ectopically expressing indicated receptors or transduced with EV and stimulated overnight ± Zym-D. Mean ± SEM from biological triplicates (left) and representative histograms (right) are plotted. Note that these data are the same data as in Fig. 6A, with bar graphs plotted here as MFI relative to untreated controls to better emphasise the response to Zym-D. Dotted line represents 1. (B) Flow cytometric analysis of surface Dectin-1 (C7) expression in RAW 264.7 cells ectopically expressing the indicated receptors or transduced with EV. Cell lines were established after sorting for equal expression of Dectin-1. (C) Flow cytometric analysis of surface DNGR-1 (C9) expression in C9 KO MuTuDCs reconstituted or not with the indicated chimeric receptors. Cell lines were established after sorting for equal expression of DNGR-1. (B, C) Representative histograms (left) and mean MFI ± SEM (right) are plotted from biological (B) quintuplets or (C) triplicates. (D) Flow cytometric analysis of surface marker expression by C9 KO MuTuDCs reconstituted or not with indicated receptors stimulated overnight ± DNGR-1L. Mean ± SEM from biological triplicates (left) and representative histograms (right) are plotted. CCR7 is shown relative to untreated controls and the dotted line represents 1. Note that these are data from the same experiments as in Fig. 6B, plotted differently. Data are representative of two (B, C), or three (A, D) independent experiments. Data were analysed using Tukey-corrected two-way ANOVA with significant values comparing against untreated samples plotted (A–D). (A) H2-D$^d$ *$P = 0.0405$, **$P = 0.0011$, ***$P = 0.0007$, ****$P < 0.0001$; CD83 *$P = 0.0374$, ***$P = 0.0001$, ****$P < 0.0001$, (D) I-A/E$^{hi}$ CD86$^{hi}$ *$P = 0.0352$ (KO/C7::C9), $P = 0.0468$ (KO/C7(G14I)::C9), ****$P < 0.0001$; CCR7 *$P = 0.0240$, ***$P = 0.0004$, ****$P < 0.0001$.

