## [Peer Review File · The EMBO Journal]

DNGR-1 signalling limits dendritic cell activation for optimal antigen cross-presentation

Michael Buck, Tomas Castro-Dopico, Oliver Schulz, Ana Cardoso, Probir Chakravarty, Nathalie Legrave, Conor Henry, Johnathan Canton, Estelle Wu, Sonia Lee, Neil Rogers, Enzo Poirier, William Stainier, Victor Bosteels, Eleanor Childs, James MacRae, Mark Skehel, Santiago Zelenay, and Caetano Reis e Sousa

Corresponding author(s): Caetano Reis e Sousa (caetano@crick.ac.uk) , Michael Buck (michael.buck@crick.ac.uk)

Review Timeline:

Submission Date:	30th Jul 25
Editorial Decision:	27th Aug 25
Revision Received:	11th Sep 25
Accepted:	28th Sep 25

Editor: Ioannis Papaioannou

Transaction Report:

This manuscript was transferred to The EMBO JOURNAL following peer review at another journal.

RESPONSE TO EDITOR

In response to the Editor's request, we have now provided source data for all figures included in the manuscript. We have also stated clearly that data correspond to biological replicates (e.g. individual mice or independently treated cell batches) in the legends for these source data and in the revised manuscript.

*Furthermore, we have added additional data to strengthen our statistical analyses. For **Figures 1M-N**, and **Extended Data Figures 1C, 1F, 1G, 2A, 2D, and 4B**, we have now combined data from multiple experiments to perform proper 2-way ANOVA statistical tests. Where biological duplicates are shown, we have plotted the data as mean \pm SD (e.g. **Figures 2E-F, 2H**). Representative plots of at least two independent experiments are shown in **Figures 2E-F**. Statistical significance in these individual experiments could not be calculated as each group included biological duplicates. It was, instead, measured by combining data from experimental repeats and is now depicted in **Extended Data Figures 2A and 2D**. All error bars are depicted throughout as specified in the figure legends. Where they are apparently missing is because the value is too small to be visible (**Figures 1A-B, 1D, 1F, 5C, 6C, 6E** and **Extended Data Figure 5B**), as stated in the Materials and Methods.*

Finally, we provide here a synopsis of the number of experiments carried out to support our conclusions:

DNGR-1 signalling induces cross-presentation of ligand-associated antigens, but does not activate cDC1

- 1. Confirmed with both BM-FLT3L cDC1 and MuTuDCs (two independent systems)*
- 2. Confirmed with two different ligand-associated antigen sources (dead cells and ligand-associated beads)*
- 3. All experiments verified independently ≥ 2 times*

A single amino acid substitution rescues the ability of DNGR-1 to activate cDC1

- 1. Confirmed with two independent cellular systems (MuTuDC and RAW264.7 cells)*
- 2. Key observation with MuTuDC data verified independently 3 times (and also verified again in Figure 6 in separate experiments)*
- 3. All other experiments verified independently ≥ 2 times except for bulk RNA-Seq, which is from one experiment with $n = 3$ per group; generally accepted given the cost*

DNGR-1 signalling does not induce a glycolytic switch in cDC1

- 1. Confirmed with two independent cellular systems (MuTuDC and RAW264.7 cells)*
- 2. Extracellular flux and activation data performed independently 2 times*
- 3. Flux experiments independently confirmed and supported by metabolomics experiment, which is from one experiment with $n = 4$ per group; generally accepted given the cost*

cDC1 activation by DNGR-1 is curtailed by SHIP1

- 1. Proteomics data set identifying SHIP1 is from one experiment with $n = 6$ per group; generally accepted given the cost*
- 2. Western blot data was performed ≥ 3 times. Densitometry data represents the combined data from independent experiments and not from technical replicates*
- 3. SHIP1 deficiency and targeting data was performed independently 2 times and also verified by inhibitor data once more in the extended data*

An activatory mutant DNGR-1 displays compromised cross-presentation activity

- 1. Key observation with cross-presentation data was performed independently 3 times*
- 2. Phagosomal rupture data and its connection with SHIP1 was performed independently 2 times*

Efficient cross-presentation is balanced against activation

- 1. Key observation with disrupting activation from Dectin-1 with DNGR-1 mutation was performed independently 3 times in both MuTuDCs and RAW264.7 cells*

2. *Observation of enhancing cross-presentation using a single bead assay was performed independently 2 times (a technically challenging and reagent-intensive experiment)*
-

RESPONSE TO REVIEWERS

We thank all the reviewers for their support, constructive comments, and suggestions. In this revised manuscript, we have included new results, re-drafted figures and made textual changes to address the reviewers' comments and improve clarity, as detailed below and (highlighted).

Reviewer #1 (Remarks to the Author):

Buck et al. nicely studied the balance of DC activation and cross presentation by two CLRs, DNGR-1/Clec9A and Dectin-1. They here demonstrate that a single amino acid is responsible for a reduced DC activation, if Clec9A was triggered. This resulted in a better antigen cross presentation by the DCs, which was intrinsic to the receptor (the induction of the receptor into a non-prof APC allowed for a certain level of cross presentation). The exchange of the defined residue allowed not only to render Dectin-1 as an also less activation-more-cross presentation CLR, but also the other way round. Although nicely presented, I have still some points/questions.

Major:

- You present nicely the (non) activation of DCs by DNGR1L and other ligands. How does this relate to antibodies to DNGR1/Clec9a for antigen targeting. Are they working the same?

*Our lab and others have previously reported that antigen-coupled-antibodies against DNGR-1 administered in vivo do not activate dendritic cells or elicit robust cytotoxic immunity in the absence of adjuvants and can even lead to the generation of regulatory T cells¹⁻⁶. As was shown in **Figure 1I**, this happens even though cross-linking with anti-DNGR-1 antibodies induces signalling from the receptor similarly to DNGR-1L.*

*To confirm these observations, we now stimulated WT or DNGR-1 KO bone marrow-derived cDC1 with two different anti-DNGR-1 antibodies (**Extended Data Figure 1E**). Neither clone induced any appreciable cDC1 activation, in line with our results using DNGR-1L stimulated cDC1.*

Additionally, some papers have shown the independence of antigen uptake and activation of the professional antigen presenting cells for CLRs and Fc receptors, such as 10.1126/science.1136080, 10.1084/jem.20160951, 10.1084/jem.194.6.769. They should be included in the discussion.

*We apologize for this oversight and **have now amended the discussion to include these references.***

- in Figures 1 and 2, axes of the graphs are very uncommon. E.g. start at 0.7 or 0 or 0.6 for relative MFIs. Could you please present "real" delta-MFIs or delta-delta-MFIs as ratios for flow cytometry data are very sensitive to measuring errors (especially, if the basic value is very low).

*We have followed the advice of this reviewer and reviewer #2's suggestion and replaced the relative MFI data presented in original **Figures 1, 2, and 6** with raw MFI values from pooled experiments or from a representative experiment. For additional transparency, we now also include the corresponding histograms and flow cytometry plots in new **Extended Data Figures 1, 2, and 6.***

*Pooling data from multiple experiments allows us to have enough biological replicates for robust statistical analysis. Relative MFI data are now only included in a few instances when repeat experiments were analysed on different cytometers. Because the raw MFI values will differ across cytometers, they cannot be pooled unless normalised to the control. We have tried to move such relative data from the main **Figures** to the **Extended Data Figures**. Finally, occasionally, we show both raw MFI in a main **Figure** and corresponding relative MFI data in an **Extended Data Figure** for comparison (e.g., **Figure 6** and **Extended Data Figure 6**), making it clear in the legend that we are showing two analyses of the same data.*

We hope that the combination of FACS plots from a representative experiment, pooled raw MFI or, in some instances, pooled relative MFI across experiments provides the reviewer with reassurance that our analysis is not confounded by measuring errors.

- Further, in Figure 2, you present now CD86^{hi} percentage - please adapt to the same representation as otherwise there is no possibility to compare with data from Figure 1. Same is true for the axes, please adapt.

*We have moved the % I-A/E^{hi} CD86^{hi} data to into new **Extended Data Figure 2** and now include CD86 raw MFI data in **Figure 2**. The relative CD86 MFI data in **Figure 1** has also now been replaced with raw MFI values.*

- you claim an equivalent expression of the variants in Figure Extended data figure 1, however, you only present histograms and they display differences (on a log scale). Please provide some bar diagrams (or better scatter blots) and some statistics.

*We thank the reviewer for their suggestion and have now included these plots in new **Extended Data Figures 2 and 6**.*

- You nicely studied this effect in cell lines (and partially in BMDCs) and with the model antigen Ova. Is this also true for other, more relevant antigens?

*To extend our data beyond OVA, we carried out new experiments with an antigen derived from influenza virus nucleocapsid protein (NP), which is relevant for immunity to that virus. Consistent with our data using OVA, KO/C9 MuTuDCs induced more IFN- γ secretion from preactivated F5 NP-specific CD8 T cells when fed NP-expressing dead cells compared to KO/C9(I6G) MuTuDCs (**Extended Data Figure 5A**). This enhancement was not due to any difference in MHC-I levels as shown by comparable presentation of pre-processed NP peptide ASNENDAM (**Extended Data Figure 5B**).*

- Is the mechanism also "active" in primary cells? Maybe the recruitment/activity of SHIP-1 can be demonstrated for murine primary DCs?

*Following this reviewer's suggestion, we attempted to demonstrate that recognition of dead cells by DNNGR-1 *in vivo* would lead to cDC1 activation in the context of SHIP1 inhibition. We already showed that the SHIP1 inhibitor 3AC recapitulated our genetic loss-of-function data (**Extended Data Figure 4C**) and 3AC has been used *in vivo* to impair SHIP1 activity when administered intraperitoneally daily⁷⁻¹⁰.*

*We injected 3AC *i.p.* (25 mg/kg) two times into WT C57BL/6J or DNNGR-1 KO (Clec9a-eGFP knock-in mice, C9^{KI-eGFP}), one day prior and the day of immunisation with UV-C killed 20x10⁶ Jurkat T cells *i.v.* labelled with Cell Proliferation Dye eF450. 12 hours later we extracted splenocytes with collagenase IV and DNase I and assessed splenic cDC1 activation by flow cytometry (**Reviewer Only Figure A**). Although we could readily detect dead cell remnants in cDC1 and upregulation of CCR7 as has been recently published (**Reviewer Only Figure B**)¹¹, 3AC treated mice had no enhanced DNNGR-1 dependent activation compared to control mice as read-out by costimulatory molecule and chemokine receptor upregulation (**Reviewer Only Figure C**). This discrepancy with our *in vitro* observations could be due to a couple reasons: 1) the amount of SHIP1 inhibitor might not be sufficient to inhibit its activity with DNNGR-1 or 2) abundant natural barriers to DNNGR-1 recognition such as secreted gelsolin might impair our ability to assess DNNGR-1 dependent activation *in vivo*.*

*As such, we cannot currently ascertain that enhanced activation via DNNGR-1 modulates primary cDC *in vivo*. We are generating mice that ectopically express DNNGR-1 vs. DNNGR-1(I6G) from the R26 locus after Cre recombinase excision of the upstream stop cassette.*

Crossing these mice to DNGR-1 knock-in Cre mice will allow us to examine the physiological consequence of replacing endogenous DNGR-1 with the activatory version of the receptor in a follow-up study.

REVIEWER ONLY FIGURE

- Could these results translated to the human system as both receptors are found there, too?

We expect our results demonstrating the trade-off between induction of activation and cross-presentation by Dectin-1 vs. DNGR-1 signalling will translate to the human system as the glycine and isoleucine residues upstream of the hemITAM in Dectin-1 and DNGR-1, respectively, are conserved. We have now included this point in the discussion.

Minor:

- please provide gating strategies, purity of the used cells as supplemental figures.

We have added these data to **Extended Data Figures 1A-B**.

- please avoid judgements, such as "faithfully", page 5

We have removed such words.

Reviewer #2 (Remarks to the Author):

The study from Buck and Reis e Sousa examines the type II transmembrane protein CLEC9A/DNGR-1 in cDC1 with respect to its role in antigen process vs. DC activation. Previous work from this lab shows DNGR-1 is a receptor for F-actin exposed on dead cells that aids in antigen capture and cross-presentation. This new study wishes to clarify a point that DNGR-1 does not activate DCs, unlike other receptors such as TLRs, and argues that this stems from recruiting SHIP1 phosphatase to the tyrosine 7 in the hemiTAM of the cytosolic tail. They show that mutating Y7 to F in a mutant Mutu DC cell will convert DNGR1 into an activating receptor with reduced the cross-presentation. The authors wish to conclude that the tail of DNGR-1 favors cross-presentation of captured antigen over DC activation and wish to suggest this might be a way to minimize inflammation in response to dead cells, although this paper did not make the Y7F mutation in vivo. The story is all fine for this journal, as it is showing work for an in vitro Cell line system, but not an in vivo validation, but it would still be a welcome clarification to the field, as the consequences of SYK activation by DNGR-1 have been unclear and potentially over-interpreted.

Paper summary. Fig. 1 argues that DNGR1 signaling is required for cross-presentation of dead cell-associated antigens but does not activate cDC1s. Lysenin index and confocal are shown to illustrate the authors model of lysosomal rupture as a mechanism. Figure 2 shows that a single amino acid substitution (I6G) in the DNGR-1 cytoplasmic domain activates cDC1 along with changes in gene expression similar to Dectin-1 signaling. In Fig. 3, the study claims that DNGR-1 signaling does not induce the glycolytic switch caused by other activating receptors, but that the I6G mutant DNGR-1 does cause this change in glycolysis and glucose metabolism. Next, the authors show that SHIP1 is recruited to DNGR-1 better than it is to I6G DNGR-1. SHIP1 deficiency rescues cDC1 activation by DNGR-1.

Figure 5 wants to claim that the 16G DNGR-1 mutant reduces the capacity for the Mutu for cross-presentation, and makes the claim that this is associated with reduced phagosomal damage. The data in Fig. 5 is not very strong data. The authors use only IFN γ production, and while they show “statistically significant” changes, the signal is still quite positive., so that it seems that there is a quantitative difference evident in the titrations but no absolute loss of activity. In Figure 6, the study makes a chimeric Dectin-1 extracellular with the ICD of DNGR-1. Making the Dectin have the Y7F reduces activation. These results suggest a trade-off between activation and cross-presentation functions, although the study is light on how these changes mechanistically cause this. The link to SHIP is at least part of the story.

Overall, the part of the study related to signaling by DNGR-1/CLEC9A are fine to report. Other parts, such as the lysenin-mCherry confocal, are a less rigorous and detract from the rigor of the study.

On the issue of rigor, there are some technical issues I would like to point out and would ask if the authors want to address them.

Error bars, axes, reproducibility and statistics.

Many of the figures and multiple panels are devoted to showing “MFI” for various surface markers. (Fig. 1K, M, N; Fig. 2E, F; Fig. 6A, B). There is not a single FACS plot to demonstrate the quality of staining. Many of the panels have the vertical axis altered to start at some particular level, not zero (e.g. 0.6 relative MFI). which is always concerning with respect to data manipulations. Further, in all of these bar graphs, there is an exceedingly small error bar. The problem is that the paper doesn't say how many individual measurements were used to obtain these extremely tight error bars for MFI, whether they are technical replicates, or different biological samples, and frankly the use of axes that start at 0.8 and go to 2.0, that then claim significance for differences that are from 1.0 to 1.5, well this is stretching the tolerance of believability.

We thank the reviewer for their summary of our work and apologize that our data presentation gave cause for concern. Reviewer #1 also expressed similar sentiments. Therefore, we have now replaced the relative MFI data presented in original **Figures 1, 2, and 6** with raw MFI values from pooled experiments or from a representative experiment. For additional transparency, we now also include the corresponding histograms and flow cytometry plots in new **Extended Data Figures 1, 2, and 6**.

Pooling data from multiple experiments allows us to have enough biological replicates for robust statistical analysis. Relative MFI data are now only included in a few instances when repeat experiments were analysed on different cytometers. Because the raw MFI values will differ across cytometers, they cannot be pooled unless normalised to the control. We have tried to move such relative data from the main **Figures** to the **Extended Data Figures**. Finally, occasionally, we show both raw MFI in a main **Figure** and corresponding relative MFI data in an **Extended Data Figure** for comparison (e.g. **Figure 6** and **Extended Data Figure 6**), making it clear in the legend that we are showing two analyses of the same data.

We hope that the combination of FACS plots from a representative experiment, pooled raw MFI or, in some instances, pooled relative MFI across experiments provides the reviewer with reassurance that our analysis is robust and believable.

Although we had indicated in our original figure legends whether data were measured in duplicate, or triplicate, etc., we have now made it explicitly clear in the revised text whether **technical or biological replicates** are represented. For BM-FLT3L cDC1 cultures, biological replicates refer to parallel independent cultures from different mice. For MuTuDC and RAW 264.7 cells, biological replicates refer to independently treated cell batches. This is now explicitly stated in the Methods.

What do the raw FACS data look like. Do the changes really seem physiologically significant, not just “statistically significant” and how many replicates does it take to have an error bar that is this small?

As mentioned above, **we have now included plots of the raw flow cytometry data and explicitly mention how many replicates are depicted**. In the main figures, representative experiments are shown and we also show combined data from independent experiments to provide robust statistical analyses. Experiments performed with biological replicates using in vitro cultures or cell lines can give very precise data. To further ensure the robustness and accuracy of our results, we have done all our stimulations across a range of dose titrations. We believe that comparison of dose response profiles adds further robustness to the differences we report among the various engineered receptors.

At least the figure legends need to say biologic replicates or technical, and if technical, how many, and how many individual different biological replications of the experiment were there, and so on.

As indicated above, **we have now made this clearer in the revised text**.

Second, the use of the lysenin-mCherry assay as a surrogate for lysosomal rupture should either be deleted or this part of the study should be substantially increased. While this aspect of cross-presentation is extraneous to the authors main claim regarding activation vs. presentation, the data shown for lysenin are not rigorous as currently provided. The lysenin index is a measure of the number of red phagosomes assayed by confocal, but the link to damage is not validated.

Previous studies from our, Felix Randow's and Johnathan Canton's lab have extensively validated lysenin-mCherry accumulation as a surrogate read-out for phagosomal damage¹²⁻¹⁴. Lysenin binds to sphingomyelin^{15,16}, a lipid primarily located in the outer leaflet of the plasma membrane. Therefore, phagosomal sphingomyelin is found mainly on the luminal side and our cytosolic probe only accumulates within phagosomes when damage makes the lumen accessible. **We have revised the text to make this point clearer.**

Additionally, **we provide confocal microscopy images in new Extended Data Figure 5C** across a consecutive series of 0.5 μm Z-slices focusing on a lysenin-mCherry⁺ phagosome that clearly show a hole in the phagosomal membrane with a diameter of 0.5 μm -1 μm . This was also shown previously for lysenin-mCherry⁺ phagosomes when analysing the ultrastructure by serial block face scanning electron microscopy¹³.

The authors should demonstrate that the rupture is leading to cytosolic antigen release.

Our previous study provides extensive evidence that phagosomal rupture leads to cytosolic antigen release¹³. We used chimeric receptors bearing a Dectin-1 extracellular domain with a DNGR-1 cytosolic domain or signalling deficient mutants and demonstrated DNGR-1 dependent phagosome-to-cytosol (P2C) antigen release using two independent assays: 1) loss of FRET signal from probe CCF4 upon zymosan-absorbed β -lactamase release into the cytosol after phagocytosis and 2) enhanced cell death after zymosan-absorbed cytochrome c entry into the cytosol from phagosomes. To connect this phenomenon to antigen release for cross-presentation, we performed the second assay with zymosan-co-absorbed with OVA and cytochrome c, titrating the time of exposure to kill only a fraction of the antigen presenting cells (APCs). Loss of cross-presentation was observed, due to loss of APCs that contain the relevant antigen compared to APCs fed with zymosan particles loaded only with OVA.

Lysenin itself binds to specific lipids and is a pore forming protein from bacteria. But the authors have not presented the evidence in this study that this is a measure of lysosomal rupture but leave it at a correlative level. It is originally a pore forming protein but this probe only retains the binding domain and does not have the pore forming domain...

*This reviewer is correct about the protein's native function and characteristics¹⁷⁻¹⁹. However, our study employs a probe consisting of a full-length oligomerization-deficient mutant (W20A) version of lysenin that has lost pore forming activity but retains the ability to bind sphingomyelin²⁰⁻²². We apologize that this was not made explicitly clear and **have amended the text for clarity.***

As mentioned above, **we have included additional evidence in images provided in Extended Figure 5C** of phagosomal rupture using this probe.

Reviewer #3 (Remarks to the Author):

In the present manuscript, entitled “DNGR-1/CLEC9A signaling limits cDC1 activation to optimize cross-presentation of dead cell antigens”, Michael Buck et al. uncover the differential signaling pathway triggered by DNGR-1 and Dectin-1 to promote antigen cross-presentation and DC activation, respectively. The authors found that, remarkably, a single amino acid difference (isoleucine vs. glycine) adjacent to the hemITAM motif in DNGR-1 prevents DC activation by activating the phosphatase SHIP1.

The authors have also performed an extensive metabolic characterization of the glycolytic pathway triggered by DNGR-1. However, while PKM2 activity, required for cDC1 activation, is not triggered upon DNGR-1 stimulation, the stimulation of PKM2 activity failed to restore cDC1 activation downstream of DNGR-1, suggesting that PKM2 activity is not sufficient to rescue DC activation. In contrast, SHIP1 deficiency rescued cDC1 activation downstream of DNGR-1 signaling.

While the mechanism linking SHIP activity to increased phagosomal damage or Ag export to the cytosol is not uncovered in the present manuscript, the authors satisfactorily discuss potential mechanisms based on published reports.

However, given the association between DNGR-1 and Ag cross-presentation, the authors could address whether MHC-I expression or the expression of immunoproteasome subunits is modified by DNGR-1 or SHIP activity (Fig. 1 shows MHC-II expression, but not MHC-I).

We thank the reviewer for their insight and constructive comments. We have included new data in Extended Data Figures 1C, 1E and 1G that show that DNGR-1 signalling does not affect MHC-I expression in BM-FLT3L cDC1 or MuTuDCs after stimulation with DNGR-1L. Additionally, our peptide controls in our antigen presentation assays featured in Figures 1A, 1D, 5C, 5K, 6E, and Extended Data Figure 5B demonstrate equal dose response curves for peptide presentation among cDC1 that are SHIP1 sufficient or deficient, or that express mutant versions of DNGR-1 (I6G or Dectin-1 tail). These peptide controls confirm that any changes in cross-presentation we observed with these modifications are not masked by differences in MHC-I levels or indeed in any other parameter that might impact DC-T cell interaction (e.g., expression of adhesion molecules).

Is SHIP activity only associated with phagosomal damage?

We do not believe that SHIP1 activity is only associated with phagosomal damage as it is well-known to act as a negative regulator of the PI3K/AKT signalling pathway. However, in the context of DNGR-1 dependent cross-presentation, we suspect that it might have a role in mediating phagosomal damage. New results, included in Figure 5I-L, show that SHIP1 KO MuTuDCs display impaired cross-presentation of DNGR-1L-coupled OVA or dead cell associated-OVA to OT-I effector T cells but are equally competent are presenting pre-processed peptide. As this defect was not due to a reduction in phagocytosis (Figure 5L), we speculate based on published reports that SHIP1 activity likely regulates phagosomal damage. We have amended the discussion section to include these points.

Is there any link between SHIP1 activity and the restriction of the glycolysis pathway mediated by DNGR-1?

We thank the reviewer for this interesting question. We stimulated DNGR-1 KO MuTuDCs or those expressing C9(I6G) or C9 that were SHIP1 sufficient or deficient with DNGR-1L and assayed changes in ECAR by extracellular flux analysis. As shown in new Extended Data Figure 4, SHIP1 deficient cells rapidly increased glycolysis, comparably to cells expressing C9(I6G). These observations are consistent with our data showing rescue of cDC1 activation in SHIP1 KO cells by DNGR-1.

Is a certain type of metabolic state required for optimal cross-presentation?

This reviewer brings up an excellent question, which follows from the previous point and is the subject of current investigation in the lab. We have added some discussion of this point to the revised text.

REFERENCES

1. Joffre, O.P., Sancho, D., Zelenay, S., Keller, A.M. & Reis e Sousa, C. Efficient and versatile manipulation of the peripheral CD4+ T-cell compartment by antigen targeting to DNGR-1/CLEC9A. *Eur J Immunol* **40**, 1255-1265 (2010).
2. Caminschi, I. *et al.* The dendritic cell subtype-restricted C-type lectin Clec9A is a target for vaccine enhancement. *Blood* **112**, 3264-3273 (2008).
3. Bonifaz, L. *et al.* Efficient targeting of protein antigen to the dendritic cell receptor DEC-205 in the steady state leads to antigen presentation on major histocompatibility complex class I products and peripheral CD8+ T cell tolerance. *J Exp Med* **196**, 1627-1638 (2002).
4. Mahnke, K., Qian, Y., Knop, J. & Enk, A.H. Induction of CD4+/CD25+ regulatory T cells by targeting of antigens to immature dendritic cells. *Blood* **101**, 4862-4869 (2003).
5. Hawiger, D., Masilamani, R.F., Bettelli, E., Kuchroo, V.K. & Nussenzweig, M.C. Immunological unresponsiveness characterized by increased expression of CD5 on peripheral T cells induced by dendritic cells in vivo. *Immunity* **20**, 695-705 (2004).
6. Kretschmer, K. *et al.* Inducing and expanding regulatory T cell populations by foreign antigen. *Nat Immunol* **6**, 1219-1227 (2005).
7. Brooks, R. *et al.* SHIP1 inhibition increases immunoregulatory capacity and triggers apoptosis of hematopoietic cancer cells. *J Immunol* **184**, 3582-3589 (2010).
8. Brooks, R. *et al.* Coordinate expansion of murine hematopoietic and mesenchymal stem cell compartments by SHIPi. *Stem Cells* **33**, 848-858 (2015).
9. Chowdhury, B.P. *et al.* SHIP1 inhibition via 3-alpha-amino-cholestane enhances protection against Leishmania infection. *Cytokine* **171**, 156373 (2023).
10. Gumbleton, M. *et al.* Dual enhancement of T and NK cell function by pulsatile inhibition of SHIP1 improves antitumor immunity and survival. *Sci Signal* **10** (2017).
11. Bosteels, V. *et al.* LXR signaling controls homeostatic dendritic cell maturation. *Sci Immunol* **8**, eadd3955 (2023).
12. Ellison, C.J., Kukulski, W., Boyle, K.B., Munro, S. & Randow, F. Transbilayer Movement of Sphingomyelin Precedes Catastrophic Breakage of Enterobacteria-Containing Vacuoles. *Curr Biol* **30**, 2974-2983 e2976 (2020).
13. Canton, J. *et al.* The receptor DNGR-1 signals for phagosomal rupture to promote cross-presentation of dead-cell-associated antigens. *Nat Immunol* **22**, 140-153 (2021).
14. Gonzales, G.A. *et al.* The pore-forming apolipoprotein APOL7C drives phagosomal rupture and antigen cross-presentation by dendritic cells. *Sci Immunol* **9**, eadn2168 (2024).
15. Sekizawa, Y., Kubo, T., Kobayashi, H., Nakajima, T. & Natori, S. Molecular cloning of cDNA for lysenin, a novel protein in the earthworm *Eisenia foetida* that causes contraction of rat vascular smooth muscle. *Gene* **191**, 97-102 (1997).
16. Yamaji, A. *et al.* Lysenin, a novel sphingomyelin-specific binding protein. *J Biol Chem* **273**, 5300-5306 (1998).
17. De Colibus, L. *et al.* Structures of lysenin reveal a shared evolutionary origin for pore-forming proteins and its mode of sphingomyelin recognition. *Structure* **20**, 1498-1507 (2012).
18. Bokori-Brown, M. *et al.* Cryo-EM structure of lysenin pore elucidates membrane insertion by an aerolysin family protein. *Nat Commun* **7**, 11293 (2016).
19. Podobnik, M. *et al.* Crystal structure of an invertebrate cytolysin pore reveals unique properties and mechanism of assembly. *Nat Commun* **7**, 11598 (2016).
20. Kiyokawa, E. *et al.* Spatial and functional heterogeneity of sphingolipid-rich membrane domains. *J Biol Chem* **280**, 24072-24084 (2005).
21. Kwiatkowska, K. *et al.* Lysenin-His, a sphingomyelin-recognizing toxin, requires tryptophan 20 for cation-selective channel assembly but not for membrane binding. *Mol Membr Biol* **24**, 121-134 (2007).

22. Kulma, M. *et al.* Sphingomyelin-rich domains are sites of lysenin oligomerization: implications for raft studies. *Biochim Biophys Acta* **1798**, 471-481 (2010).

Dear Caetano,

Thank you again for transferring your revised manuscript (EMBOJ-2025-122024-T) to The EMBO Journal for our consideration, and for your patience during peer review. Your revised manuscript has been sent back to two of the three original referees who had previously assessed the first version of your manuscript at another journal, and we have now received their comments, which you can find below. For your information, I would like to clarify that the remaining referee was also contacted but did not agree to re-review the manuscript.

I am very pleased to say that, as you will see, both referees #1 and #3 are satisfied with the revision, explaining that all initially raised major concerns have been adequately addressed. There is only one minor remaining request by referee #1 regarding the need to avoid over-interpretation of the results, which should be addressed in the Discussion and the title in a final version of your manuscript. Please upload this version to our online submission system along with a brief point-by-point response to the referee's comments detailing all changes to the manuscript.

There are also a few changes and corrections from the editorial side we kindly request you to address in the final version of your manuscript, before we can move forward with its formal acceptance and publication:

- The manuscript should be uploaded as a Word file without Figures (only their legends should remain in this file).
- Main and EV Figures should be uploaded as individual, high-resolution Figure files.
- Please provide a list of up to 5 keywords (preferably broad terms to enhance the online search engine discoverability of your article) after the Abstract of your revised manuscript.
- Please change heading "MAIN" to "Introduction".
- The References list must be alphabetical, with only the names of the first 10 co-authors of each citation provided and followed by "et al.". Please refer to our guide to authors for more information on our References format:
<https://www.embopress.org/page/journal/14602075/authorguide#referencesformat>.
- Before submitting your revision, all mass spectrometry and RNA-sequencing datasets (and computer code, if applicable) produced in this study need to be deposited in appropriate public databases (see <https://www.embopress.org/page/journal/14602075/authorguide#dataavailability>). The accession numbers, databases, and the specific URLs to the datasets should be listed in a formal "Data availability" section (placed after Methods), following the example: "The RNA-seq datasets produced in this study are available in the following database: Gene Expression Omnibus GSE46843 (<https://www.ncbi.nlm.nih.gov/geo/query/acc.cgi?acc=GSE46843>)". Please note that all links should resolve to a page where the data can be accessed, and that the Data Availability Section is restricted to new primary data that are part of this study.
- Heading "Declaration of interests" should be renamed to "Disclosure and competing interests statement".
- The author contributions statement should be removed from the manuscript file. Instead, we use CRediT to specify the contributions of each author in the journal submission system. Please feel free to use the free text box to provide more detailed descriptions during submission. See also our guide to authors for more information:
<https://www.embopress.org/page/journal/14602075/authorguide#authorshipguidelines>.
- The callouts for Expanded View (EV) Figures should be updated to "Figure EV1-EV6" throughout the manuscript.
- Please upload along with your revised manuscript a completed author checklist, which you can download from our guide to authors (<https://www.embopress.org/page/journal/14602075/authorguide>). Please note that this checklist will also be part of the Peer Review File.
- The materials and methods need to be described in the manuscript using our structured methods format, which is now required for all research articles. According to this format, the Methods section (formerly "Materials and Methods") includes a single "Reagents and Tools Table" -listing key reagents, experimental models, software and relevant equipment including their sources and relevant identifiers- followed by a "Methods and Protocols" section describing the methods. Please download and fill our Reagents and Tools Table template (.docx), which you can find in our author guide:
<https://www.embopress.org/page/journal/14602075/authorguide#structuredmethods>. When submitting your revised manuscript, please do not include the Reagents and Tools Table in the Methods section of the manuscript but instead upload it as a separate file choosing the file type "Reagent Table".
- Expanded View (EV) Figure labels, callouts and legends should be renamed to "Figure EV1-EV6" instead of "Extended Data

Figure 1-6".

- Please note that EMBO press papers are accompanied online by:

A) a short (2 sentences) summary of the findings and their significance,

B) 2-5 short bullet points highlighting the key results, and

C) a synopsis image in .jpg or .png format that is exactly 550 pixels wide and 300-600 pixels high (the height is variable). Please note that all text needs to be legible at the final size.

Please upload this information along with your revised manuscript (the text for A and B should be provided in a separate Word file).

- During our routine data checks, our data editors have raised the following queries regarding data, figures, and legends. Please make sure that the requests below are completely addressed in the final version of your manuscript (please highlight all changes in the revised manuscript):

1. We noticed that $n=2$ in Figures 2F, H. In such cases, the individual data points must be shown in the Figures, and no statistics can be shown or discussed.

2. Please provide the exact p-values in the legends of Figures 1A, D, E, H K, L, M, N; 4E, 5A B, G, J; 6A, B, E; EDF 1C, EDF 2A, D; EDF 3C, D; EDF 4B-E; EDF 5A, EDF 6A, D.

3. Please indicate the statistical test used for data analysis in the legends of Figure EDF 1C.

- The order of manuscript sections and their headings must be corrected as follows: Title page - Abstract and Keywords - Introduction - Results - Discussion - Methods - Data Availability - Acknowledgements - Disclosure and Competing Interests Statement - References - Figure Legends - main Tables (Table 1 in this case) - Expanded View Figure Legends.

Please also note that as part of the EMBO publications' Transparent Editorial Process, The EMBO Journal publishes online a Peer Review File along with each accepted manuscript. This File will be published in conjunction with your paper and will include the referee reports, your point-by-point response and all pertinent correspondence relating to the manuscript. You can opt out of this by letting the editorial office know (contact@embojournal.org). If you do opt out, the Peer Review File link will point to the following statement: "No Peer Review File is available with this article, as the authors have chosen not to make the review process public in this case."

We look forward to seeing a final version of your manuscript as soon as possible. Please let us know if you have any questions and use this link to submit your revision: <https://emboj.msubmit.net/cgi-bin/main.plex>.

Best regards,

Ioannis

Referee #1:

The manuscript of Buck et al., studying the balance of DC activation and cross presentation by two CLR, namely DNNGR-1/Clec9A and Dectin-1, demonstrates that a single amino acid is sufficient for a reduced DC activation, if Clec9A was triggered. This led to a better antigen cross presentation by the DCs, which was intrinsic to the receptor.

The authors further expanded and solidified the manuscript according to my initial concerns (also partly shared by other reviewers).

I think that in the current version, all major points have been clarified.

As, with the tools currently available to the authors, the effects could, unfortunately, not be shown for primary isolated cells or directly in vivo, I would suggest to add this in the title to not over-interpret the results for the community and to "leave space" for

upcoming studies (as proposed by the authors) looking more closely into the in vivo / primary cDC1 relevance.

I would propose "DNGR-1/CLEC9A signalling limits DC activation to optimise cross-presentation of dead cell antigens by BMDCs and MutuDCs" or similar.

Referee #3:

The authors have adequately addressed the reviewer's comments.

RESPONSE TO REVIEWERS

We thank all the reviewers for their support, constructive comments, and suggestions. In this revised manuscript, we have made textual changes to address the reviewers' comments and improve clarity, as detailed below.

Referee #1:

The manuscript of Buck et al., studying the balance of DC activation and cross presentation by two CLRs, namely DNGR-1/Clec9A and Dectin-1, demonstrates that a single amino acid is sufficient for a reduced DC activation, if Clec9A was triggered. This led to a better antigen cross presentation by the DCs, which was intrinsic to the receptor.

The authors further expanded and solidified the manuscript according to my initial concerns (also partly shared by other reviewers). I think that in the current version, all major points have been clarified.

As, with the tools currently available to the authors, the effects could, unfortunately, not be shown for primary isolated cells or directly in vivo, I would suggest to add this in the title to not over-interpret the results for the community and to "leave space" for upcoming studies (as proposed by the authors) looking more closely into the in vivo / primary cDC1 relevance. I would propose "DNGR-1/CLEC9A signalling limits DC activation to optimise cross-presentation of dead cell antigens by BMDCs and MuTuDCs" or similar.

We thank the reviewer for their positive feedback on our revised manuscript. To caution the over interpretation of our results as the reviewer suggests, we have added a sentence at the end of our discussion stating that the physiological consequences of modulating the ability of DNGR-1 to balance activation versus cross-presentation remains to be tested in vivo. With regards to the suggested title change, in order to avoid confusion over the ambiguous term BMDCs and the unfamiliarity of MuTuDCs to the broader readership of the journal, we have chosen to modify the title of the manuscript to refer to "dendritic cells" instead to provide a more nuanced title.

Referee #3:

The authors have adequately addressed the reviewer's comments.

We thank the reviewer for their positive feedback on our revised manuscript.

Dear Caetano,

Congratulations on an excellent manuscript! I am very pleased to inform you that it has been accepted for publication in The EMBO Journal. Thank you for comprehensively addressing the initially raised referee concerns and the editorial requests for corrections and reformatting.

If you have any questions, please do not hesitate to contact the Editorial Office. Thank you for your contribution to The EMBO Journal. Working with you has been a pleasure!

Best regards,

Ioannis
